# ADAPTIVE HL-GAUSSIAN: A VALUE FUNCTION LEARNING METHOD WITH DYNAMIC SUPPORT ADJUSTMENT

## ABSTRACT

Recent research indicates that using cross-entropy (CE) loss for value function learning surpasses traditional mean squared error (MSE) loss in performance and scalability, with the HL-Gaussian method showing notably strong results. However, this method requires a pre-specified support for representing the categorical distribution of the value function, and an inappropriately chosen interval for the support may not match the time-varying value function, potentially impeding the learning process. To address this issue, we theoretically establish that HL-Gaussian inherently introduces a projection error during the learning of the value function, which is dependent on the support interval. We further prove that an ideal interval should be sufficiently broad to reduce truncation-induced projection errors, yet not so excessive as to counterproductively amplify them. Guided by these findings, we introduce the Adaptive HL-Gaussian (AHL-Gaussian) approach. This approach starts with a confined support interval and dynamically adjusts its range by minimizing the projection error. This ensures that the interval's size stabilizes to adapt to the learning value functions without further expansion. We integrate AHL-Gaussian into several classic value-based algorithms and evaluate it on Atari 2600 games and Gym Mujoco. The results show that AHL-Gaussian significantly outperforms the vanilla baselines and standard HL-Gaussian with a static interval across the majority of tasks.

## 1 Introduction

Deep Reinforcement Learning (DRL) has achieved significant success across various practical applications (Badia et al., 2020; Shah et al., 2022; Fawzi et al., 2022; Degrave et al., 2022; OpenAI, 2022), among which value-based methods (Mnih et al., 2015; Silver et al., 2017) are the most widely adopted frameworks. Within this framework, value functions are typically approximated using neural networks and learned to fit the Bellman targets. A common approach for this is to employ mean squared error (MSE) as the regression objective (Mnih et al., 2015; Haarnoja et al., 2018; Fujimoto et al., 2018a). Alternatively, a class of methods (Bellemare et al., 2017; Dabney et al., 2018) models the value function as a categorical distribution on a finite support, capturing its distributional properties, and employs cross-entropy (CE) loss for learning as a classification objective. Recently, Farebrother et al. (2024) reviewed three typical approaches within this paradigm and revealed that using CE loss rather than MSE loss can significantly improve the training performance and exhibit scaling law as model complexity increases. In particular, Farebrother et al. (2024) found that HL-Gaussian (Imani & White, 2018), a specialized method among the three, which involves projecting Bellman target scalar into a categorical distribution derived from Gaussian distribution, can produce the most remarkable results. These empirical benefits have been attributed to several hypotheses, including improved gradient stability (Ehsan Imani, 2024), better feature representation (Zhang et al., 2023), implicit biases (Stewart et al., 2023), and greater resilience to noisy targets and non-stationary environments (Farebrother et al., 2024).

However, HL-Gaussian and analogous approaches that represent value functions through categorical distributions share a fundamental limitation: they necessitate a predetermined interval for the support, $[v_{\min}, v_{\max}]$, within which the value functions must be confined. Intuitively, the choice of

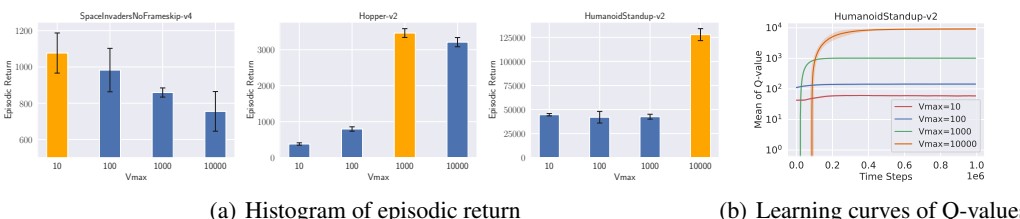

(a) Histogram of episodic return       (b) Learning curves of Q-values

Figure 1: Comparison of different interval magnitudes

the interval significantly impacts the training performance: on one hand, value function is used to characterize the actual returns generated when a policy is executed in a task. However, an inappropriate interval can restrict the value function to an unreasonable range, making it difficult to capture task-specific return information and thus affecting the final performance. As demonstrated in Figure 1(a), the magnitude of the optimal interval range varies across three distinct tasks, reflecting the task-specific nature of the interval. On the other hand, the value function dynamically changes and often exhibits numerical growth during the learning process. Consequently, a static and overly narrow interval may fail to adapt to the temporal variability of the value function, thereby negatively suppressing its upward trend. As depicted in Figure 1(b), the value functions, each induced by different intervals, all exhibit an upward trajectory. However, their ultimate convergence values are significantly influenced by the range of the interval.

Therefore, for HL-Gaussian to be effective in RL and achieve widespread adoption, the current static method of setting intervals, which relies on task-specific priors, is inadequate. A more promising approach lies in developing a dynamic support adjustment mechanism that is both task-agnostic and value-function-aware. This advancement would enable HL-Gaussian to reach its full potential across a wide range of real-world applications.

In this study, we explore the influence of the support interval $[v_{\min}, v_{\max}]$ on the learning of the value function within the HL-Gaussian approach. Initially, we establish that HL-Gaussian introduces a projection error during value function fitting, arising from the truncation of the Bellman target within $[v_{\min}, v_{\max}]$ and its discretization into categorical distributions. Subsequently, we demonstrate the relationship between these two types of errors and $[v_{\min}, v_{\max}]$: when the Bellman target is within $[v_{\min}, v_{\max}]$ and away from the boundaries, the overall projection error is minimal. Otherwise, the truncation error is substantial. Additionally, the projection error rises linearly with the expansion of $[v_{\min}, v_{\max}]$. Therefore, we conclude that the ideal interval must be wide enough to minimize truncation error but not so broad as to incur counterproductive projection error. Leveraging these theoretical insights, we propose an approach that begins with a confined interval and dynamically adjusts its range by optimizing the projection error. This method is task-agnostic and can adapt the interval dynamically based on the learning value function. We term this approach Adaptive HL-Gaussian (AHL-Gaussian).

We integrate AHL-Gaussian with several classic value-based algorithms, including DQN (Mnih et al., 2015), SAC (Haarnoja et al., 2018), and TD3 (Fujimoto et al., 2018b), and apply it to tasks with both discrete action spaces (Atari 2600 games) and continuous action spaces (Gym Mujoco). The results consistently show that across the majority of tasks, AHL-Gaussian, without relying on any prior knowledge of the tasks, greatly improves the performance of the original algorithms it is applied to, as well as the HL-Gaussian method that is specially fine-tune for each task. This performance is attributed to three key characteristics of AHL-Gaussian: (i) the universality of its mechanism, which is both task-agnostic and value function-aware; (ii) the modularity that facilitates flexible integration as a plug-in within a variety of algorithms; and (iii) the insensitivity to its remaining parameters associated with HL-Gaussian, thus enhancing robustness.

## 2 Preliminaries

**Reinforcement Learning (RL)**. We consider the reinforcement learning (RL) problem where an agent interacts with the environment by selecting an action $a_t \in \mathcal{A}$ in the current state $s_t \in \mathcal{S}$. Afterward, the agent receives a reward $r_{t+1} \in \mathbb{R}$ and transitions to the next state $s_{t+1} \in \mathcal{S}$ according

to the environment's transition model $\mathcal{P}(\cdot|s_t, a_t)$. The return is defined as the cumulative discounted sum of rewards: $G_t = \sum_{k=0}^{\infty} \gamma^k r_{t+k+1}$, where $\gamma \in [0, 1)$ is the discount factor. The agent's objective is to learn a policy $\pi : \mathcal{S} \rightarrow \mathcal{P}(\mathcal{A})$ that maximizes the expected return. The action-value function, $Q^\pi(s, a) = \mathbb{E}_\pi [G_t \mid s_t = s, a_t = a]$, represents the expected return from taking action $a$ in state $s$ and following policy $\pi$ thereafter. The optimal action-value function is $Q^*(s, a) = \mathbb{E}_{\pi^*} [G_t \mid s_t = s, a_t = a]$.

Q-learning and actor-critic methods are two widely used approaches within the value-based algorithm framework. Q-learning directly learns the optimal action-value function $Q^*$, which satisfies the optimal Bellman equation (1). On the other hand, actor-critic algorithms focus on learning the action-value function $Q^\pi$ corresponding to the current policy $\pi$, which satisfies the standard Bellman equation (2).

$$Q^*(s_t, a_t) = r_{t+1} + \gamma \mathbb{E}_{s_{t+1} \sim \mathcal{P}(\cdot|s_t, a_t)} \max_{a'} Q^*(s_{t+1}, a') := (\mathcal{T}^* Q^*)(s_t, a_t). \tag{1}$$

$$Q^\pi(s_t, a_t) = r_{t+1} + \gamma \mathbb{E}_{s_{t+1} \sim \mathcal{P}(\cdot|s_t, a_t), a_{t+1} \sim \pi(\cdot|s_{t+1})} Q^\pi(s_{t+1}, a_{t+1}) := (\mathcal{T}^\pi Q^\pi)(s_t, a_t). \tag{2}$$

Let $\widehat{\mathcal{T}}$ denote the approximation of either $\mathcal{T}^*$ or $\mathcal{T}^\pi$, depending on the specific algorithm. $Q$ function is typically learned by minimizing the temporal difference (TD) error between $Q(s_t, a_t)$ and the Bellman target $\widehat{\mathcal{T}} Q(s_t, a_t)$ for all $(s_t, a_t)$ in the replay buffer $\mathcal{D}$, with a mean squared error (MSE) as objective:

$$\mathcal{L}_{\text{MSE}} = \mathbb{E}_{\mathcal{D}} \left( \widehat{\mathcal{T}} Q(s_t, a_t) - Q(s_t, a_t) \right)^2.$$

**Cross-entropy Loss in RL.** Distributional RL methods (Bellemare et al., 2017; Dabney et al., 2018) and recent work (Farebrother et al., 2024) propose representing value function as a categorical distribution and replacing MSE with cross-entropy (CE) loss in value function learning. Specifically, $Q(s, a)$ is represented as the expected value of a random variable $Z(s, a)$, and $Z(s, a)$ obeys a categorical distribution on a set of $m$ discrete locations $[z_1, \cdots, z_m]$ within $[v_{\min}, v_{\max}]$. This distribution is parameterized by the learned probabilities $\hat{p}_i(s_t, a_t)$ corresponding to each location $z_i$, which are computed from logits $l_i(s_t, a_t)$ via the softmax function. In summary:

$$Q(s_t, a_t) = \mathbb{E}[Z(s_t, a_t)], \quad \hat{p}_i(s_t, a_t) = \frac{\exp(l_i(s_t, a_t))}{\sum_{j=1}^{m} \exp(l_j(s_t, a_t))}. \tag{3}$$

It is essential that the Bellman target is also represented as a categorical distribution, supported at the same locations. Let $p_i(s_t, a_t)$ denote the probability associated with $z_i$, ensuring that the equation $\sum_{i=1}^{m} p_i(s_t, a_t) z_i = \left( \widehat{\mathcal{T}} Q \right)(s_t, a_t)$ holds true. Consequently, the CE loss for learning $\hat{p}_i(s_t, a_t)$ is defined as

$$\mathcal{L}_{\text{CE}} = \mathbb{E}_{\mathcal{D}} \left[ -\sum_{i=1}^{m} p_i(s_t, a_t) \log \hat{p}_i(s_t, a_t) \right]. \tag{4}$$

**HL-Gaussian in RL** For constructing the target categorical distributions $[p_1(\cdot), \cdots, p_m(\cdot)]$, Farebrother et al. (2024) reviews various strategies and identifies that HL-Gaussian (Imani & White, 2018; Ehsan Imani, 2024) delivers the best performance. Specifically, assume the interval $[v_{\min}, v_{\max}]$ is uniformly divided into $m$ bins, each with a width $w$, where the center of each bin is $z_i$. Consider a truncated Gaussian distribution with variance $\sigma^2$, centered at the Bellman target value $\mu = \left( \widehat{\mathcal{T}} Q \right)(s_t, a_t)$. The probability density function $f(y)$ is given by:

$$f(y) = \frac{1}{Z\sigma\sqrt{2\pi}} e^{-\frac{(y-\mu)^2}{2\sigma^2}}, \quad Z = \frac{1}{2} \left( \text{erf}\left( \frac{v_{\max} - \mu}{\sqrt{2}\sigma} \right) - \text{erf}\left( \frac{v_{\min} - \mu}{\sqrt{2}\sigma} \right) \right). \tag{5}$$

Then, the probability assigned to each center $z_i$ is defined as:

$$p_i(s_t, a_t) = \frac{1}{2Z} \left( \text{erf}\left( \frac{z_i + \frac{w}{2} - \mu}{\sqrt{2}\sigma} \right) - \text{erf}\left( \frac{z_i - \frac{w}{2} - \mu}{\sqrt{2}\sigma} \right) \right). \tag{6}$$

The CE loss, which employs histogram densities derived from equation (6) as the target categorical distribution, is referred to as HL-Gaussian. Indeed, HL-Gaussian demonstrates superior optimization characteristics over traditional regression techniques. This assertion is corroborated by the research of Ehsan Imani (2024), which reveals that the local Lipschitz constant, or the gradient norm of HL-Gaussian, is significantly smaller than that of MSE at each iteration. Such a trait is highly advantageous for the optimization process, as highlighted by Hardt et al. (2016).

# 3 Method

## 3.1 Projection Error of HL-Gaussian

While HL-Gaussian exhibits desirable optimization traits, when projecting the Bellman target $\widehat{\mathcal{T}}Q(s_t, a_t)$ onto the support interval and representing it with a categorical distribution, a projection error is inevitably introduced. We define this as:

$$\mathcal{E}_{v_{\min},v_{\max},m,\sigma}(s_t, a_t) = \sum_{i=1}^{m} p_i(s_t, a_t) z_i - \left(\widehat{\mathcal{T}}Q\right)(s_t, a_t). \tag{7}$$

**Proposition 3.1.**

$$\mathcal{L}_{\mathrm{MSE}} \leq 4 \max(|v_{\min}|, |v_{\max}|)^2 \mathcal{L}_{\mathrm{CE}} + \mathbb{E}_{\mathcal{D}}\mathcal{E}_{v_{\min},v_{\max},m,\sigma}^2 + C, \tag{8}$$

*where $C$ is a constant independent of the learning functions $[\hat{p}_1(\cdot), \cdots, \hat{p}_m(\cdot)]$.*

Given that the projection error $\mathcal{E}_{v_{\min},v_{\max},m,\sigma}$ is insignificant for every $(s_t, a_t)$ pair within $\mathcal{D}$, Proposition 3.1 posits that utilizing HL-Gaussian to minimize $\mathcal{L}_{\mathrm{CE}}$ is an effective strategy for optimizing the traditional TD error $\mathcal{L}_{\mathrm{MSE}}$. Furthermore, as highlighted by Ehsan Imani (2024), the CE loss holds a theoretical edge over the MSE loss in the optimization process, facilitating a more efficient path to the optimal solution with a reduced number of gradient steps—a concept supported by a wealth of empirical data (Farebrother et al., 2024; Ehsan Imani, 2024). Consequently, the adoption of $\mathcal{L}_{\mathrm{CE}}$ as the optimization objective is well-founded in both theoretical understanding and practical results, which justifies our focus on $\mathcal{L}_{\mathrm{CE}}$ in this study.

Nonetheless, if the projection error $\mathcal{E}_{v_{\min},v_{\max},m,\sigma}$ is significant, the TD error $\mathcal{L}_{\mathrm{MSE}}$ may remain substantial even with $\mathcal{L}_{\mathrm{CE}}$ minimized, owing to the lingering impact of $\mathcal{E}_{v_{\min},v_{\max},m,\sigma}$. According to the error propagation theory in Approximate Policy/Value Iteration (Farahmand et al., 2010), the accumulation of one-step TD-errors can significantly impair the optimality of the final value function, thereby leading to a suboptimal policy. Consequently, effectively reducing the projection error is of paramount importance.

## 3.2 Relationship between Projection Error and Support Interval

In this section, we delve into the origins of the projection error and examine the interplay between $\mathcal{E}_{v_{\min},v_{\max},m,\sigma}$ and the interval $[v_{\min}, v_{\max}]$. This intrinsic connection will serve as the cornerstone to create a mechanism that is both task-agnostic and value function-aware, designed to adjust the support interval effectively.

We begin by introducing some notations. Given a Bellman target $\mu = \widehat{\mathcal{T}}Q(s, a)$, let $m_0$ represent the center of the bin that contains $\mu$, and define $\delta := \mu - m_0$ [1]. It is straightforwardly that $0 \leq |\delta| \leq \frac{w}{2}$ and the value will vary depending on the specific position of $\mu$. Define $F_{\mu,\sigma}(x, y)$ as the cumulative probability of the Gaussian distribution $\mathcal{N}(\mu, \sigma^2)$ on interval $[x, y]$. Further define:

$$h = \lfloor \frac{\min\left(|v_{\max} - \mu|, |\mu - v_{\min}|\right)}{w} \rfloor, \quad k = \lfloor \frac{\max\left(|v_{\max} - \mu|, |\mu - v_{\min}|\right)}{w} \rfloor.$$

As illustrated in Figure 2, $h$ represents the number of bins between $\mu$ and the closer boundary of $[v_{\min}, v_{\max}]$, while $k$ represents the number of bins between $\mu$ and the farther boundary of $[v_{\min}, v_{\max}]$.

**Theorem 3.1.** *With the number of bins $m$ fixed, let $w = \beta\sigma$, where $\beta$ is a hyperparameter, and let $Z$ be as defined in (5). Then, for a wide ragne of $\beta$,*

$$\mathcal{E}_{a,b,m,\sigma}(s, a) = \mathcal{E}_{discretization} + \mathcal{E}_{truncation},$$

*with*

$$\mathcal{E}_{discretization} = \delta \cdot \left(\mathbf{1}_{\{v_{\min} \leq \mu < v_{\max}\}} \frac{F_{0,1}(-\beta h, \beta h) + o(1)}{2Z} - 1\right),$$

$$(h+1)w \frac{F_{0,1}(-\beta(k+1), -\beta(h+1))}{2Z} \leq |\mathcal{E}_{truncation}| \leq kw \frac{F_{0,1}(-\beta k, -\beta h)}{2Z},$$

*where $o(1)$ represents a constant far less than 1.*

---

[1] To keep the expression concise, we assume that bins also exist outside the interval $[v_{\min}, v_{\max}]$.

**Remark 1.** *It is important to note that the assumption of fixed $m$ and the setting of $w = \beta\sigma$ aligns with the configuration in Farebrother et al. (2024), ensuring that the number of non-zero elements in the vector $[p_1(s,a), \cdots, p_m(s,a)]$ remains consistent. This stability preserves the representational capacity throughout the process.*

Theorem 3.1 posits that the projection error can be dissected into two components: truncation error and discretization error. The truncation error materializes when the target is confined within the interval $[v_{\min}, v_{\max}]$, whereas the discretization error arises from employing the categorical distribution to depict the Bellman target. The former has a direct linear relationship with $w$, and the latter mirrors a linear association with $\delta$, with the proportionality constants being exclusively derived from $\beta, h, k$. Note that $w$ signifies the interval span with a fixed number of bins $m$, thus, a pivotal insight is unveiled: the overall projection error ascends in direct proportion to the enlargement of the interval span.

We will offer more refined estimates for each of the two errors.

**Theorem 3.2.** *With the number of bins $m$ fixed, let $w = \beta\sigma$, where $\beta$ is a hyperparameter that can be selected over a wide range. According to the relationship between $\mu$ and $[v_{min}, v_{max}]$, we have :*

*(i) If $\mu \in (v_{min}, v_{max})$,*

$$|\mathcal{E}_{discretization}| = C_{\beta,1} \cdot \left((he^{h^2})^{-1} + o(1)\right) \cdot |\delta|, \quad |\mathcal{E}_{truncation}| = C_{\beta,m,2} \cdot (he^{h^2})^{-1} \cdot w,$$

*(ii) If $\mu \in (v_{min}, v_{max})$ and $h = 0$, or if $\mu \notin (v_{min}, v_{max})$,*

$$|\mathcal{E}_{discretization}| = |\delta|, \quad |\mathcal{E}_{truncation}| \geq C_{\beta,3}(h+1)w,$$

*where $C_{\beta,1}$, $C_{\beta,m,2}$ and $C_{\beta,3}$ are constants dependent only on the hyperparameters and $o(1)$ is a constant far less than 1.*

Note that Theorem 3.2 confirms the idea that the projection error increases linearly with $w$, consistent with Theorem 3.1. Moreover, with $w$ fixed, Theorem 3.2 further shows how the projection error varies as $\mu$ changes, as illustrated in Figure 2. Specifically, when $\mu$ is well within the interval $[v_{\min}, v_{\max}]$ and far from the boundaries (indicating a larger $h$ in case (i)), both types of errors decrease exponentially with an increase in $h$, resulting in a minimal overall pro-

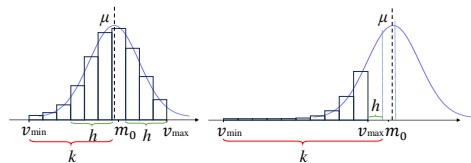

Figure 2: Illustrations of Case (i) and (ii)

jection error. On the other hand, as $\mu$ approaches the boundaries or falls outside $[v_{\min}, v_{\max}]$ (indicating case (ii)), the discretization error stays at $|\delta|$ while the truncation error becomes predominant, growing linearly with $h$, and thus significantly increasing the overall projection error compared to the initial case.

Building upon the insights from both Theorem 3.1 and 3.2, we summarize our key finding as follows.

> **Key Finding :** (i) Given the fixed interval $[v_{\min}, v_{\max}]$, the projection error remains minimal if $\mu$ is positioned within $[v_{\min}, v_{\max}]$ and is sufficiently distant from either boundary. In contrast, the projection error increases markedly if $\mu$ nears either boundary of $[v_{\min}, v_{\max}]$ or if $\mu$ lies outside this interval. (ii) As the interval range $[v_{\min}, v_{\max}]$ widens, the projection error increases linearly.

## 3.3 Adaptive HL-Gaussian Method

Proposition 3.1 emphasizes the importance of reducing projection error. Furthermore, the key finding from the previous section suggests that an ideal support interval should be broad enough to encompass all Bellman targets comfortably within it, thereby reducing truncation error, yet not so excessive as to induce counterproductive projection error. This inspires us to develop a dynamic interval adjustment mechanism by optimizing the projection error.

Specifically, we introduce a learnable variable $\xi$, and let $[-\xi, \xi]$ represent the current support interval. This interval yields a dynamic bin width $w_\xi = 2\xi/m$. Besides, let $\sigma_\xi = \alpha w_\xi$ with

the ratio $\alpha = 1/\beta$ being a fixed hyperparameter. For any $(s, a)$, we project its corresponding target $\mu = \widehat{T}Q(s, a)$ onto the categorical distribution of discrete locations $[z_{1,\xi}, \cdots, z_{m,\xi}]$, where $z_{i,\xi}$ is the center of each bin associated with $[-\xi, \xi]$. This yields the projected target value $\sum_{i=1}^{m} p_{i,\xi}(s, a)z_{i,\xi}$, where $p_{i,\xi}(s, a)$ is computed by

$$p_{i,\xi}(s, a) = \frac{1}{2Z_\xi} \left( \text{erf} \left( \frac{z_{i,\xi} + \frac{w_\xi}{2} - \mu}{\sqrt{2}\sigma_\xi} \right) - \text{erf} \left( \frac{z_{i,\xi} - \frac{w_\xi}{2} - \mu}{\sqrt{2}\sigma_\xi} \right) \right) \tag{9}$$

$$Z_\xi = \frac{1}{2} \left( \text{erf} \left( \frac{\xi - \mu}{\sqrt{2}\sigma_\xi} \right) - \text{erf} \left( \frac{-\xi - \mu}{\sqrt{2}\sigma_\xi} \right) \right).$$

We use the projection error $\mathcal{L}_{\text{projection}}(\xi)$ to measure whether current interval $[-\xi, \xi]$ is suitable or not to project all the Bellman targets in $\mathcal{D}$ and minimize $\mathcal{L}_{\text{projection}}(\xi)$ to obtain an appropriate $\xi$:

$$\min_\xi \mathcal{L}_{\text{projection}}(\xi) := \mathbb{E}_\mathcal{D} \left( \sum_{i=1}^{m} p_{i,\xi}(s_t, a_t)z_{i,\xi} - \left( \widehat{\mathcal{T}}Q \right)(s_t, a_t) \right)^2. \tag{10}$$

HL-Gaussian with dynamic support interval adjustment is defined as *Adaptive **HL**-Gaussian* (AHL-Gaussian). The procedure for updating the value function once can be outlined in Algorithm 1.

---

**Algorithm 1** Value Function Update with AHL-Gaussian

---

Fix hyperparameters $\beta$, $m$. The parameterized logits function is $[l_1^\theta, \cdots, l_m^\theta]$ and $[l_1^{\bar{\theta}}, \cdots, l_m^{\bar{\theta}}]$. The bound is $\xi$.

1: Sample a random minibatch $\mathcal{B}$ of transitions from replay memory $\mathcal{D}$;
2: **for** $i = 1$ to $|\mathcal{B}|$ **do**
3:     For transition $(s_i, a_i, r_i, s_{i+1})$, calculate $[\hat{p}_{1,\xi}(s_i, a_i), \cdots, \hat{p}_{m,\xi}(s_i, a_i)]$ through (3);
4:     For transition $(s_i, a_i, r_i, s_{i+1})$, calculate Q-values $Q_{\bar{\theta}}(s_{i+1}, a')$ through (3), where $a'$ is sampled from greedy policy or current $\pi$ according to the underlying algorithm;
5:     Calculate the Bellman target value $y_i = r_i + \gamma Q_{\bar{\theta}}(s_{i+1}, a')$;
6:     Project $y_i$ into categorical distribution $[p_{1,\xi}(s_{i+1}, a'), \cdots, p_{m,\xi}(s_{i+1}, a')]$ through (9);
7: **end for**
8: Calculate $\mathcal{L}_{\text{CE}}$ on $\mathcal{B}$ by (4) and perform a gradient descent step to update $\theta$;
9: Calculate $\mathcal{L}_{\text{projection}}(\xi)$ on $\mathcal{B}$ by (10) and perform a gradient descent step to update $\xi$.

---

In practice, for tasks where the value function undergoes large changes, we suggest adding a bias term to the interval calculations, which results in the shifted interval $[-\xi + v_{\text{mean}}, \xi + v_{\text{mean}}]$, where $v_{\text{mean}}$ is the mean of the current Q-values. This ensures that the center of the interval moves in sync with the value function, thus effectively preventing the interval from becoming overly broad.

This dynamic interval adjustment mechanism does not requires any prior knowledge of the task at hand, and can automatically calibrate the interval to suit the learning value function: starting from an initially constrained interval, when the Bellman targets dynamically increases and exceeds the boundaries of the interval, the resulting projection error will drive an increase in $\xi$. Conversely, when the Bellman targets stabilizes, $\xi$ will also converges at a state that is sufficient to encompass all Bellman targets without the impetus to continue expanding. At this point, the interval has achieved a balance that is neither too large nor too small. Additionally, this method involves optimizing just a single variable, making it computationally efficient and resulting in minimal extra computational cost.

# 4 Experimental Evaluation

In this section, we undertake an empirical analysis to explore several critical questions: (i) Is the projection error introduced by AHL-Gaussian consistent with the behavioral patterns our theory anticipates? (ii) Can AHL-Gaussian be seamlessly incorporated into conventional value-based algorithms to enhance performance? (iii) Is it possible to realize the essence of AHL-Gaussian without resorting to learning-based approaches? (iv) How resilient is AHL-Gaussian in the face of variations in other hyperparameters associated with HL-Gaussian?

## 4.1 Observation Study of Projection Error

Consider the interval $[-500, 500]$. Figure 3(a) demonstrates how the projection error varies as the Bellman target $\mu$ shifts. It is evident that the error exhibits two distinct patterns based on $\mu$'s relative position to the interval. The error is minimal when $\mu$ is comfortably within $[-500, 500]$. However, the error climbs as $\mu$ nears the interval's boundaries. Once $\mu$ exceeds the interval, the error increases linearly with the distance from the boundaries. This aligns perfectly with our primary finding (i), indicating that a sufficiently large support interval should be chosen to cover the majority of targets.

As we progress, we examine the patterns of projection error across a range of $\xi$ value. Figure 3 (b) illustrates the error curves for instances when $\mu$ shifts within a tighter interval $[-2\sigma_\xi, 2\sigma_\xi]$, a subset of $[-\xi, \xi]$. Here, the projection error exhibits periodic fluctuations that align with the characteristics of $\delta$. Moreover, as $\xi$ increases, the peaks of the projection error also increase in a linear fashion. Figure 3 (c) shows the scenario where $\mu$ shits within $[-1.3\xi, 1.3\xi]$. Similarly, the peak error values for each curve rise linearly with $\xi$. These empirical findings consistently support the conclusion that, in both scenarios, the error peaks climb linearly with $\xi$, thus confirming our key discovery (ii).

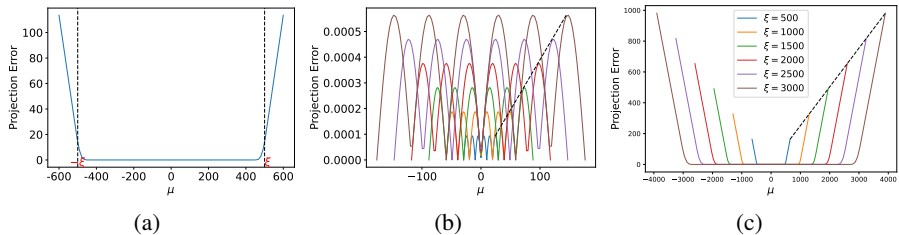

|  (a)  |  (b)  |  (c)  |

Figure 3: Panel (a) presents the projection error curve, varying with $\mu$ while keeping $\xi$ constant. Panels (b) and (c) illustrate the projection error curves across a range of $\xi$ values, under conditions where $\mu$ is either within or exceeds the limits of $[-\xi, \xi]$, respectively.

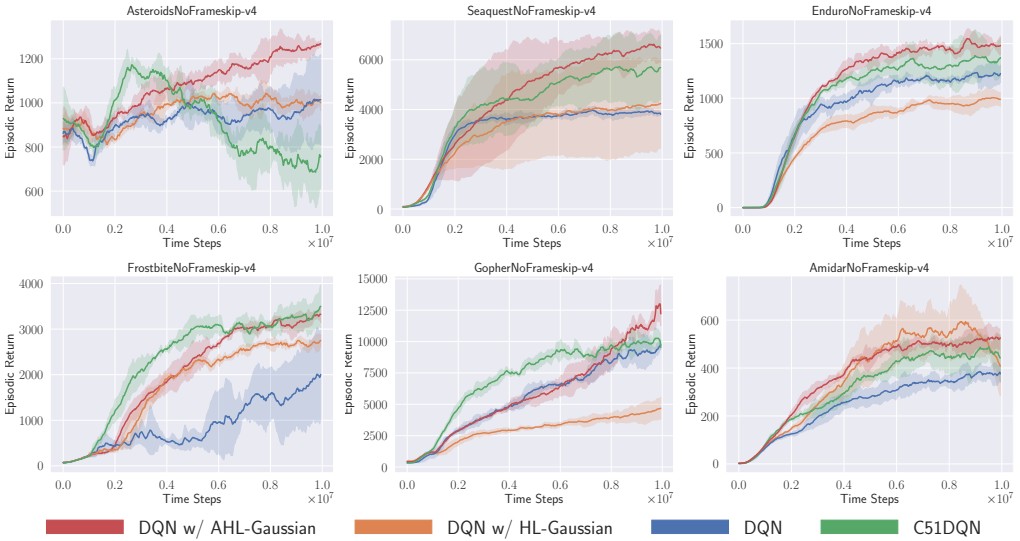

Figure 4: Performance of DQN with AHL-Gaussian.

## 4.2 Performance of AHL-Gaussian

**Integration with Q-learning Method.** We first assess the efficacy of AHL-Gaussian by integrating it with DQN (Mnih et al., 2015) and evaluate its performance on Atari 2600 games (Mnih et al., 2013). The baselines compared include the standard DQN, DQN with the conventional HL-Gaussian using a default interval of $[-10, 10]$, and the representative distributional RL method C51 (Bellemare et al., 2017). As depicted in Figure 4, DQN integrated with AHL-Gaussian excels in five out

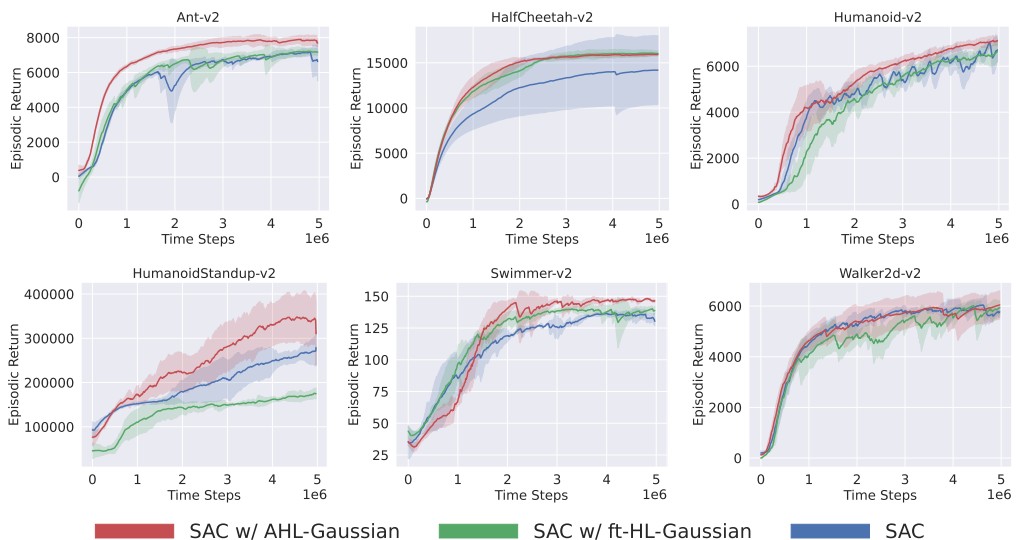

Figure 5: Performance of SAC with AHL-Gaussian.

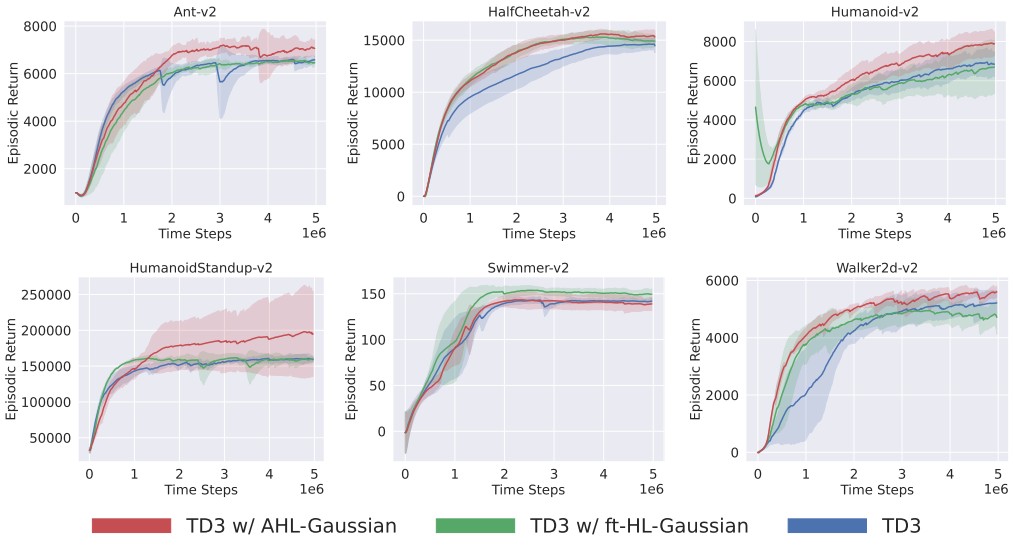

Figure 6: Performance of TD3 with AHL-Gaussian.

of six tasks, achieving significant improvements in four of them. In contrast, HL-Gaussian has resulted in reduced performance for certain tasks, demonstrating that the default static interval does indeed negatively affect the training process. This observation further validates the superiority of our dynamically adjusted mechanism.

**Integration with Actor-Critic Method.** Moreover, we have also incorporated AHL-Gaussian into the typical actor-critic algorithms SAC (Haarnoja et al., 2018) and TD3 (Fujimoto et al., 2018b), and evaluated their performance in the Gym MuJoCo environments (Todorov et al., 2012). Our baselines include not only the original SAC and TD3 but also a specially fine-tuned version of HL-Gaussian, with a customized support interval for each task, denoted as ft-HL-Gaussian. This fine-tuning was essential due to the substantial differences in return scales across various MuJoCo tasks, which made identifying a universal interval that could perform optimally across all tasks difficult.

Figures 5 and 6 demonstrate that across nearly all tested tasks, the algorithm enhanced with AHL-Gaussian surpasses both the conventional algorithms and those augmented with ft-HL-Gaussian. Moreover, in over half of these tasks, the performance advantage is substantial. In line with the previous integration with DQN, HL-Gaussian results in a performance decline in certain tasks, high-

lighting the difficulty of manually adjusting the optimal interval and the challenge a static interval faces in accommodating the fluctuations of the value function. This reaffirms the distinct advantages of AHL-Gaussian, which is both task-agnostic and value function-aware.

## 4.3 Comparative Study of AHL-Gaussian and Non-learning-based Strategies

In this section, we explore the possibility of implementing AHL-Gaussian without relying on learning mechanisms. A naive strategy would be to set $\xi$ as the maximum value of all current Bellman targets, multiplied by a coefficient $\eta$. This approach, while straightforward, aims to dynamically adapt the interval in response to fluctuations in the value function. However, as shown in Figure 7, for both Ant-v2 and Hopper-v2 tasks, setting $\eta$ to 1 results in a constrained interval range, which in turn, triggers substantial projection errors and inferior performance. Upon increasing $\eta$ to 1.1, we observe a significant improvement in Ant-v2's performance, suggesting that $\eta$ correlates well with the escalating trend of the value function. Conversely, on Hopper-v2, this adjustment causes an unwarranted surge in the value function, leading to significant projection errors and subpar performance again. This observation implies that the coefficient $\eta$, being a hyperparameter, is inherently task-specific and thus lacks the universal applicability that AHL-Gaussian offers across various tasks.

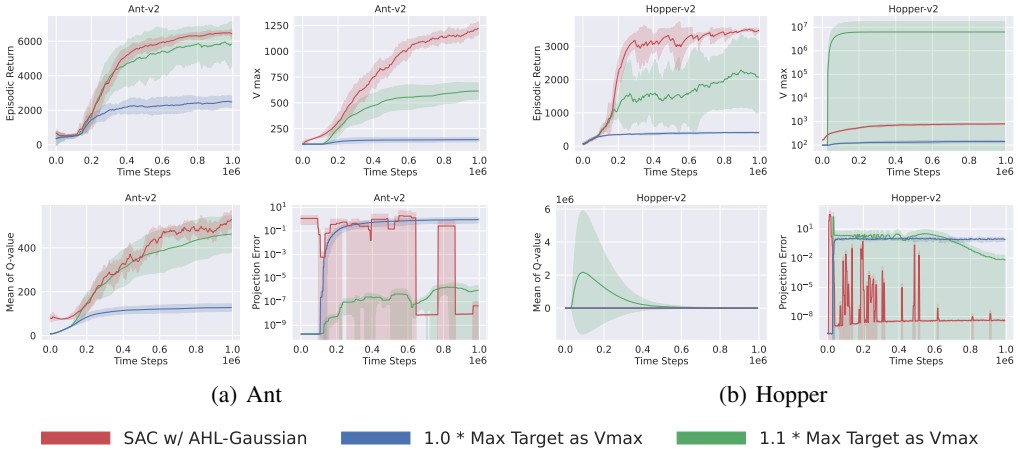

(a) Ant    (b) Hopper

**SAC w/ AHL-Gaussian**    **1.0 * Max Target as Vmax**    **1.1 * Max Target as Vmax**

Figure 7: Comparison of AHL-Gaussian to a Non-learning-based Method.

## 4.4 Robustness of AHL-Gaussian

In this section, we assess the robustness of AHL-Gaussian with respect to other involved hyperparameters. Complete experimental results are deferred in the Appendix.

**Number of Bins** ($m$). We analyzed how AHL-Gaussian performs with different values of $m$, ranging from $[11, 31, 51, 71, 91]$. Figure 8 shows that AHL-Gaussian generally holds up well regardless of $m$, but there is a slight dip in performance on a few tasks when $m$ is set too low. Given these findings, we picked the value of $m = 51$, which provides a good balance of performance and computational efficiency. This choice also aligns with the recommendations from Bellemare et al. (2017); Farebrother et al. (2024).

**Ratio of bin width to variance** ($\alpha$). We analyzed how AHL-Gaussian performs with different values of $\alpha$, ranging from $[0.5, 0.75, 1.5, 2.0, 3.0]$. Overall, AHL-Gaussian maintains reliable performance regardless of the $\alpha$ value, with only occasional performance drops on a few tasks for specific $\alpha$ choices. This aligns well with our theoretical findings, which show that Theorems 3.1 and 3.2 are valid across a wide spectrum of $\alpha$ values. We have settled on $\alpha = 1.5$ as the algorithm's hyperparameter, a choice that works well for the majority of the tasks.

**Interval update frequency.** To determine how the frequency of interval updates affects performance, we conducted a series of experiments with varying ratios of interval update frequency to

value function update frequency, as illustrated in Figure 10. The results indicate that AHL-Gaussian is quite resilient to changes in these ratios. This resilience is a practical advantage, as it allows AHL-Gaussian to maintain its performance while conserving computational resources.

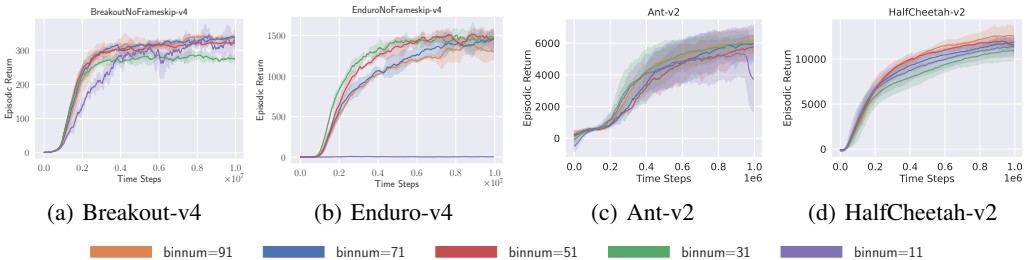

(a) Breakout-v4  (b) Enduro-v4  (c) Ant-v2  (d) HalfCheetah-v2

Figure 8: Ablation for the number of bins $m$.

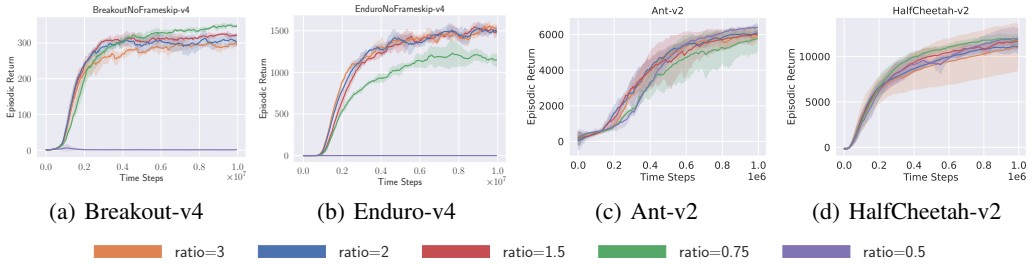

(a) Breakout-v4  (b) Enduro-v4  (c) Ant-v2  (d) HalfCheetah-v2

Figure 9: Ablation for the ratio of width to variance.

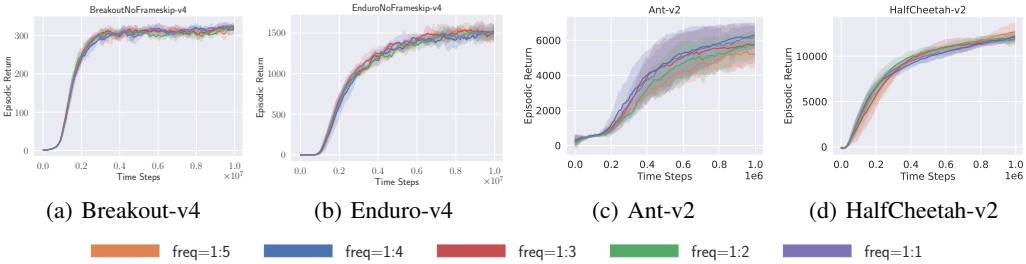

(a) Breakout-v4  (b) Enduro-v4  (c) Ant-v2  (d) HalfCheetah-v2

Figure 10: Ablation for the interval update frequency.

# 5  Conclusion and Future Work

In this paper, we concentrate on value function learning methods that leverage HL-Gaussian. We demonstrate that a misalignment between the support interval and the value function can result in substantial projection errors, which in turn can compromise the optimality of the resulting policy. Our analysis further reveals that an ideal interval should be sufficiently broad to reduce truncation-induced projection errors, yet not so extensive as to paradoxically amplify them. Motivated by these findings, we introduce AHL-Gaussian, a novel dynamic interval adjustment mechanism designed to align with the dynamic evolution of the value function. Empirical results indicate that AHL-Gaussian is compatible with a range of algorithms and can consistently boost performance across both discrete and continuous control tasks.

In our future work, we intend to broaden the application of the AHL-Gaussian approach to encompass more complex tasks. Furthermore, we aim to integrate it with a range of other distributional RL methods. We are also committed to investigating the adherence of AHL-Gaussian to a scaling law as the model's complexity increases.

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

# A Related Work

**Distributional RL** Distributional Reinforcement Learning (Distributional RL) marks a key advancement in reinforcement learning by modeling the distribution of returns instead of the expected return. The C51 algorithm (Bellemare et al., 2017) models the value function using a categorical distribution instead of a scalar and adopts cross entropy loss to learn the value function, yielding improved performance, especially in stochastic environments. Following this, QR-DQN (Dabney et al., 2018) introduces a quantile-based approximation, learning specific quantiles via the quantile Huber loss, thereby offering finer control over the distribution tails. DERL (Rowland et al., 2019) further advances DRL with expectile regression, enabling the modeling of conditional value at risk (CVaR), which is particularly advantageous for risk-averse applications. On the theoretical front, Wang et al. (2023) provides small-loss bounds for distributional RL, offering stronger convergence guarantees, while Rowland et al. (2023) extends QR-DQN through quantile temporal-difference learning, showing its statistical benefits in environments with skewed or heavy-tailed rewards. Distributional RL has also been extended to continuous action spaces through the D4PG (Barth-Maron et al., 2018) and DSAC (Duan et al., 2022), and also demonstrates the scalability and generalization in offline Q-learning across diverse multi-task data (Kumar et al., 2023). The advantage of quantization in distributional RL are also discussed (Bellemare et al., 2017) that it can better handle approximation errors, reduce chattering caused by policy updates, and mitigate state aliasing, thus improving training stability. Additionally, the distribution itself provides a rich set of predictions, allowing the agent to learn from multiple predictions rather than solely focusing on an expected value. Moreover, the distributional perspective introduces a more natural inductive bias framework for reinforcement learning, enabling the imposition of assumptions on the domain or the learning problem itself.

Our proposed AHL-Gaussian method falls within the realm of distributional RL. Yet, it distinguishes itself from the methods previously discussed by employing HL-Gaussian to project the Bellman target's value onto a discrete distribution when crafting the categorical distribution of the Bellman target. Building upon the existing limitations of HL-Gaussian, we have further introduced a mechanism for dynamic interval adjustment, which significantly differentiates AHL-Gaussian from current distributional RL methods.

**HL-Gaussian in RL** HL-Gaussian is a specialized learning method that utilizes the cross-entropy loss and constructs a target categorical distribution derived from Gaussian histogram densities. Initially proposed for regression tasks by Imani & White (2018) and Ehsan Imani (2024), it was found to primarily enhance optimization processes. Farebrother et al. (2024) later applied HL-Gaussian to reinforcement learning (RL), demonstrating significant improvements in training performance and a beneficial scaling effect as the model complexity increases. This pioneering work spurred further exploration. Denis Tarasov (2024) investigated HL-Gaussian in offline RL settings, finding it capable of delivering state-of-the-art results, albeit with occasional fluctuations. Josiah P. Hanna & Harish (2024) applied it to stochastic policy gradient RL, achieving enhanced data efficiency and stability, particularly in continuous control scenarios. Additionally, Yang Zhang (2024) successfully adapted a discrete regression method akin to HL-Gaussian for multi-agent systems.

While these studies have straightforwardly integrated HL-Gaussian with existing RL methods, they overlook a critical aspect of RL algorithms: the target function for fitting is in constant flux. Consequently, a static support interval is inadequate for fully realizing HL-Gaussian's potential in RL. To surmount this challenge, we introduced a dynamic interval adjustment mechanism, which we have both theoretically and empirically proven to be effective and universally applicable.

# B Proofs

## B.1 Proof of Proposition 3.1

**Lemma B.1** (Ehsan Imani (2024)). *Assuming that a data point $\mu$'s target distribution is $p_\mu$. Let an $m$-dimensional vector $h_x$ be a model's prediction distribution and has supports bounded by the*

*range $[a, b]$, then*

$$\left(E_{p_\mu}[z] - E_{h_x}[z(x)]\right)^2 \leq 4 \max(|a|, |b|)^2 \min\left(\frac{1}{2} D_{KL}\left(p_\mu || h_x\right), 1 - e^{-D_{KL}(p_\mu || h_x)}\right)$$

*Proof of Proposition 3.1.* Given a state-action pair $(s, a)$ in $\mathcal{D}$, let $x := (s, a)$ and its Q-value $Q(x)$ be the expectation of random variable $z(x)$, where $z(x)$ obeys the categorical distribution $h_x$. So $Q(s, a) = \mathbb{E}_{h_x}[z(x)]$ and $h_x$ is actually the model's prediction $[\hat{p}_1(s, a), \cdots, \hat{p}_m(s, a)]$ in AHL-Gaussian. Further denote the target distribution $p_\mu$ as the categorical distribution $[p_1(s, a), \cdots, p_m(s, a)]$ induced by projection function (6). Since $h_x$ has supports in the range $[v_{\min}, v_{\max}]$, we can apply Lemma B.1 to obtain that

$$[Q(s, a) - (\widehat{\mathcal{T}}Q)(s, a)]^2$$

$$= [Q(s, a) - \sum_{i=1}^m p_i(s, a) z_i + \sum_{i=1}^m p_i(s, a) z_i - (\widehat{\mathcal{T}}Q)(s, a)]^2$$

$$\leq [Q(s, a) - \sum_{i=1}^m p_i(s, a) z_i]^2 + \mathcal{E}_{v_{\min}, v_{\max}, m, \sigma}^2(s, a)$$

$$= \left(\mathbb{E}_{h_x}[z(x)] - \mathbb{E}_{p_\mu}[z]\right)^2 + \mathcal{E}_{v_{\min}, v_{\max}, m, \sigma}^2$$

$$\leq 4 \max(|v_{\min}|, |v_{\max}|)^2 \left(\frac{1}{2} D_{KL}\left(p_\mu || h_x\right)\right) + \mathcal{E}_{v_{\min}, v_{\max}, m, \sigma}^2(s, a)$$

$$= 2 \max(|v_{\min}|, |v_{\max}|)^2 \left(HL(p_\mu, h_x) - H(p_\mu)\right) + \mathcal{E}_{v_{\min}, v_{\max}, m, \sigma}^2(s, a).$$

Because $H(p_\mu)$ only depends on $p_\mu$ which is independent of the learned variable $x$, the aim is to minimize the first term: the cross-entropy between $p_\mu$ and $h_x$, we have

$$[Q(s, a) - (\mathcal{T}Q)(s, a)]^2 \leq 2 \max(|v_{\min}|, |v_{\max}|)^2 \left(HL(p_\mu, h_x)\right) + \mathcal{E}_{v_{\min}, v_{\max}, m, \sigma}^2(s, a) + C. \tag{11}$$

By taking average in $\mathcal{D}$, Proposition 3.1 can be derived straightforwardly. $\square$

## B.2 Proofs of Theorem 3.1 and Theorem 3.2

**Lemma B.2** (Burden & Faires (2010)). *Assuming there are $n$ equally spaced bins on the interval $[b_l, b_r]$, we use the sum of the function values at the midpoints of each bin multiplied by the bin width to approximate the integral $\int_{b_l}^{b_r} g(x)dx$. Then the approximation error is:*

$$\mathcal{E}_n = \frac{(b_r - b_l)^3}{24n^2} g''(\xi), \tag{12}$$

*where $\xi \in [b_l, b_r]$.*

**Lemma B.3.**

$$2 \sum_{i=1}^h \beta \left(f_{0,1}((i - \frac{1}{2})\beta)\right) = F_{0,1}(-\beta h, \beta h) + o(1). \tag{13}$$

*Proof.* For the interval $[(i-1)\beta, i\beta]$, we apply Lemma B.2 on this interval with $n = 1$, then

$$\int_{(i-1)\beta}^{i\beta} f_{0,1}(x)dx = \beta \cdot f_{0,1}((i - \frac{1}{2})\beta) + \mathcal{E}_{i,\beta}$$

$$= \beta \cdot f_{0,1}((i + \frac{1}{2})\beta) + \frac{\beta^3}{24} f_{0,1}''(\xi_{\beta,i}) \tag{14}$$

where $\xi_{\beta,i} \in [(i-1)\beta, i\beta]$. Therefore,

$$
\begin{aligned}
F_{0,1}(-\beta h, \beta h) &= 2 \sum_{i=1}^{h} \int_{(i-1)\beta}^{i\beta} f_{0,1}(x) dx \\
&= 2 \sum_{i=1}^{h} \left( f_{0,1}((i - \frac{1}{2})\beta) + \frac{\beta^3}{24} f_{0,1}''(\xi_{\beta,i}) \right) \\
&= 2 \sum_{i=1}^{h} \beta \left( f_{0,1}((i - \frac{1}{2})\beta) \right) + \sum_{i=1}^{h} \frac{\beta^3}{12} f_{0,1}''(\xi_{\beta,i})
\end{aligned}
\tag{15}
$$

In particular, given that $\beta = 1$,

$$
F_{0,1}(-h, h) = 2 \sum_{i=1}^{h} \left( f_{0,1}((i - \frac{1}{2})) \right) + \sum_{i=1}^{h} \frac{1}{12} f_{0,1}''(\xi_{1,i}).
$$

Note that $f_{0,1}''(x) = \frac{x^2-1}{\sqrt{2\pi}} e^{\frac{-x^2}{2}}$, which is $o(1)$ on $(4, \infty)$. Besides, $f_{0,1}''(x)$ is upper bounded on $[0, 4]$, thus we further define $C_i = \frac{\max_{x \in [(i-1)\beta, i\beta]} f_{0,1}''(x)}{f_{0,1}''(\xi_{1,i})}$ for $i \in [1, 4]$. This implies that

$$
\begin{aligned}
\sum_{i=1}^{h} \frac{\beta^3}{24} f_{0,1}''(\xi_{\beta,i}) &= \sum_{i=1}^{4} \frac{\beta^3}{24} f_{0,1}''(\xi_{\beta,i}) + o(1) \\
&\leq \sum_{i=1}^{4} \frac{\beta^3}{24} \max_{x \in [(i-1)\beta, i\beta]} f_{0,1}''(x) + o(1) \\
&\leq \sum_{i=1}^{4} \frac{\beta^3}{24} C_i f_{0,1}''(\xi_{1,i}) + o(1) \\
&\leq C \left( \sum_{i=1}^{4} \frac{\beta^3}{24} f_{0,1}''(\xi_{1,i}) \right) + o(1).
\end{aligned}
\tag{16}
$$

It can be empirically verified that $\left( \sum_{i=1}^{4} \frac{\beta^3}{24} f_{0,1}''(\xi_{1,i}) \right)$ is a constant and its value is $o(1)$, thus Lemma B.3 holds true directly for a wide range of $\beta$. $\qquad\square$

*Proof of Theorem 3.1.* ] We follow the notation defined in Theorem 2. Note that each bin center $z_j$ corresponds to an $m_i := m_0 + iw$. We also denote the range of bin $i$ as $\mathcal{S}_i = [m_i - \frac{w}{2}, m_i + \frac{w}{2})$. According to the relationship between $\mu$ and $[v_{\min}, v_{\max}]$, there are two cases to be considered: (i) $v_{\min} < \mu < v_{\max}$ and (ii) $\mu \geq v_{\max}$ or $\mu \leq v_{\min}$. We will assume that $\mu$ is closer to $v_{\max}$ without loss of generality and discuss the following two cases separately.

• **Case (i)** $v_{\min} < \mu < v_{\max}$.
At this situation, $h \geq 0$. We first consider the case of $h \geq 1$.

Since $v_{\max}$ is closer to $\mu$, it is directly that $h = \lfloor \frac{v_{\max}-\mu}{w} \rfloor$, $k = \lfloor \frac{\mu-v_{\min}}{w} \rfloor$, and $k \geq h \geq 1$. Besides, $v_{\max} \in \mathcal{S}_h$, and $v_{\min} \in \mathcal{S}_{-k}$.

$$\mathcal{E}_{v_{\min},v_{\max},m,\sigma} = \sum_{j=1}^{m} z_i p_i - \mu$$

$$= \frac{1}{F_{\mu,\sigma}(v_{\min},v_{\max})} \sum_{i=-k}^{h} m_i \int_{\mathcal{S}_i} f(z)dz - \mu$$

$$= \frac{1}{F_{\mu,\sigma}(v_{\min},v_{\max})} \sum_{i=-k}^{h} (m_0 + iw) \int_{\mathcal{S}_i} f(z)dz - \mu$$

$$= \frac{1}{F_{\mu,\sigma}(v_{\min},v_{\max})} m_0 \sum_{i=-k}^{h} \int_{\mathcal{S}_i} f(z)dz + \frac{1}{F_{\mu,\sigma}(v_{\min},v_{\max})} w \sum_{i=-k}^{h} i \int_{\mathcal{S}_i} f(z)dz - \mu. \tag{17}$$

Recall that intervals $\mathcal{S}_i$ are disjoint and $\bigcup_{i=-k}^{h} \mathcal{S}_i = [v_{\min}, v_{\max}]$. So the series in the first term becomes $F_{\mu,\sigma}(v_{\min}, jinv_{\max})$. Therefore

$$(17) = (m_0 - \mu) + \frac{1}{F_{\mu,\sigma}(v_{\min},v_{\max})} w \left( \sum_{i=-k}^{-1} i \int_{\mathcal{S}_i} f(z)dz + \sum_{i=1}^{h} i \int_{\mathcal{S}_i} f(z)dz \right)$$

$$= -\delta + \frac{1}{F_{\mu,\sigma}(v_{\min},v_{\max})} w \left( \sum_{i=-k}^{-1} i F_{\mu,\sigma}\left(m_0 + iw - \frac{w}{2}, m_0 + iw + \frac{w}{2}\right) \right)$$

$$+ \sum_{i=1}^{h} i F_{\mu,\sigma}\left(m_0 + iw - \frac{w}{2}, m_0 + iw + \frac{w}{2}\right)$$

$$= \frac{1}{F_{\mu,\sigma}(v_{\min},v_{\max})} w \left( \sum_{i=1}^{k} (-i) F_{\mu,\sigma}\left(m_0 - iw - \frac{w}{2}, m_0 - iw + \frac{w}{2}\right) \right)$$

$$\left( -\delta + \sum_{i=1}^{h} i F_{\mu,\sigma}\left(m_0 + iw - \frac{w}{2}, m_0 + iw + \frac{w}{2}\right) \right)$$

$$= \left( -\delta + \frac{1}{F_{\mu,\sigma}(v_{\min},v_{\max})} w \sum_{i=1}^{h} i \left( F_{\mu,\sigma}\left(m_0 + iw - \frac{w}{2}, m_0 + iw + \frac{w}{2}\right) - F_{\mu,\sigma}\left(m_0 - iw - \frac{w}{2}, m_0 - iw + \frac{w}{2}\right) \right) \right)$$

$$+ \left( \frac{1}{F_{\mu,\sigma}(v_{\min},v_{\max})} w \sum_{i=h+1}^{k} \left( -F_{\mu,\sigma}\left(m_0 - iw - \frac{w}{2}, m_0 - iw + \frac{w}{2}\right) \right) \right) \tag{18}$$

$$:= \mathcal{E}_{\text{discretization}} + \mathcal{E}_{\text{truncation}}. \tag{19}$$

Due to the symmetry of Gaussian distribution, $F_{\mu,\sigma}(v_{\min},v_{\max}) = F_{\mu,\sigma}(2\mu - v_{\max}, 2\mu - v_{\min})$. We define $a_i := iw - \frac{w}{2}$ and $b_i := iw + \frac{w}{2}$, and the second series in (18) becomes

$$\sum_{i=1}^{h} i \left( F_{\mu,\sigma} \left( m_0 + a_i, m_0 + b_i \right) - F_{\mu,\sigma} \left( m_0 - b_i, m_0 - a_i \right) \right)$$

$$= \sum_{i=1}^{h} i \left( F_{\mu,\sigma} \left( m_0 + a_i, m_0 + b_i \right) - F_{\mu,\sigma} \left( 2\mu - m_0 + a_i, 2\mu - m_0 + b_i \right) \right)$$

$$= \sum_{i=1}^{h} i \left( F_{\mu,\sigma} \left( \mu - \delta + a_i, \mu - \delta + b_i \right) - F_{\mu,\sigma} \left( \mu + \delta + a_i, \mu + \delta + b_i \right) \right)$$

$$= \sum_{i=1}^{h} i \left( \Phi_\mu \left( \mu - \delta + b_i \right) - \Phi_\mu \left( \mu - \delta + a_i \right) + \Phi_\mu \left( \mu + \delta + a_i \right) - \Phi_\mu \left( \mu + \delta + b_i \right) \right)$$

$$= \sum_{i=1}^{h} i \left( -F_{\mu,\sigma} \left( \mu + b_i - \delta, \mu + b_i + \delta \right) + F_{\mu,\sigma} \left( \mu + a_i - \delta, \mu + a_i + \delta \right) \right)$$

$$= -\sum_{i=1}^{h} i F_{\mu,\sigma} \left( \mu + b_i - \delta, \mu + b_i + \delta \right) + \sum_{i=1}^{h} i F_{\mu,\sigma} \left( \mu + a_i - \delta, \mu + a_i + \delta \right) \qquad (20)$$

Since $iw + \frac{w}{2} = (i+1)w - \frac{w}{2}$, we can replace $b_i$ by $a_{i+1}$ and have

$$(20) = -\sum_{i=1}^{h} i F_{\mu,\sigma} \left( \mu + a_{i+1} - \delta, \mu + a_{i+1} + \delta \right) + \sum_{i=1}^{h} i F_{\mu,\sigma} \left( \mu + a_i - \delta, \mu + a_i + \delta \right)$$

$$= -\sum_{i=2}^{h} (i-1) F_{\mu,\sigma} \left( \mu + a_i - \delta, \mu + a_i + \delta \right) + \sum_{i=1}^{h} i F_{\mu,\sigma} \left( \mu + a_i - \delta, \mu + a_i + \delta \right)$$

$$= \sum_{i=2}^{h} F_{\mu,\sigma} \left( \mu + a_i - \delta, \mu + a_i + \delta \right) + F_{\mu,\sigma} \left( \mu + a_1 - \delta, \mu + a_1 + \delta \right)$$

$$= \sum_{i=1}^{h} F_{\mu,\sigma} \left( \mu + a_i - \delta, \mu + a_i + \delta \right) = \sum_{i=1}^{h} F_{0,\sigma} \left( a_i - \delta, a_i + \delta \right)$$

$$= \sum_{i=1}^{h} F_{0,\sigma} \left( iw - \frac{w}{2} - \delta, iw - \frac{w}{2} + \delta \right)$$

Given that $|\delta| \le \frac{w}{2} = \frac{\beta}{2}\sigma$, then $\frac{|\delta|}{\sigma} \le 1$ for $\beta \le 2$, thus we can use the first-order Taylor approximation with

$$\sum_{i=1}^{h} F_{0,\sigma} \left( iw - \frac{w}{2} - \delta, iw - \frac{w}{2} + \delta \right) = 2\delta \sum_{i=1}^{h} f_{0,\sigma}(iw - w/2) + o(1)$$

So the second term in (18) becomes

$$\delta \cdot \frac{1}{F_{\mu,\sigma}(v_{\min}, v_{\max})} \left( 2w \sum_{i=1}^{h} f_{0,\sigma}(iw - w/2) + o(1) \right)$$

$$= \delta \cdot \frac{1}{F_{\mu,\sigma}(v_{\min}, v_{\max})} \left( 2\beta \sum_{i=1}^{h} f_{0,1}(i\beta - \beta/2) + o(1) \right)$$

$$= \delta \cdot \frac{1}{2Z} \left( F_{0,1}(-\beta h, \beta h) + o(1) \right). \qquad (21)$$

where (21) comes from Lemma B.3.

Now we bound the third series in (18) as follows:

$$\sum_{i=h+1}^{k} (-iw)F_{\mu,\sigma}\left(m_0 - iw - \frac{w}{2}, m_0 - iw + \frac{w}{2}\right)$$

$$\leq -(h+1)wF_{\mu,\sigma}\left(m_0 - kw - \frac{w}{2}, m_0 - (h+1)w + \frac{w}{2}\right)$$

$$\leq -(h+1)wF_{\mu,\sigma}\left(\mu + \delta - kw - \frac{w}{2}, \mu + \delta - (h+1)w + \frac{w}{2}\right)$$

$$= -(h+1)wF_{0,\sigma}\left(\delta - kw - \frac{w}{2}, \delta - (h+1)w + \frac{w}{2}\right)$$

$$\leq -(h+1)wF_{0,\sigma}\left(-(k+1)w, -(h+1)w\right)$$

$$= -(h+1)wF_{0,1}\left(-\beta(k+1), -\beta(h+1)\right). \tag{22}$$

Similarly,

$$\sum_{i=h+1}^{k} (-iw)F_{\mu,\sigma}\left(m_0 - iw - \frac{w}{2}, m_0 - iw + \frac{w}{2}\right)$$

$$\geq -kwF_{\mu,\sigma}\left(m_0 - kw - \frac{w}{2}, m_0 - (h+1)w + \frac{w}{2}\right)$$

$$= -kwF_{\mu,\sigma}\left(\mu + \delta - kw - \frac{w}{2}, \mu + \delta - (h+1)w + \frac{w}{2}\right)$$

$$= -kwF_{0,\sigma}\left(\delta - kw - \frac{w}{2}, \delta - (h+1)w + \frac{w}{2}\right)$$

$$\geq -kwF_{0,\sigma}\left(-kw, -hw\right)$$

$$= -kwF_{0,1}\left(-\beta k, -\beta h\right). \tag{23}$$

Combining (21), (22) and (23), we can obtain that

$$\mathcal{E}_{\text{discretization}} = \delta \cdot \left(\frac{F_{0,1}(-\beta h, \beta h) + o(1)}{2Z} - 1\right),$$

$$-kw\frac{F_{0,1}\left(-\beta k, -\beta h\right)}{2Z} \leq \mathcal{E}_{\text{truncation}} \leq -(h+1)w\frac{F_{0,1}\left(-\beta(k+1), -\beta(h+1)\right)}{2Z}. \tag{24}$$

This constitutes the conclusion corresponding to case (i) in Lemma (3.1).

Next we make similar analysis to the case of $h = 0$.

$$\mathcal{E}_{a,b,m,\sigma} = \sum_{j=1}^{m} z_i p_i - \mu$$

$$= \frac{1}{F_{\mu,\sigma}(v_{\min}, v_{\max})} \sum_{i=-k}^{0} m_i \int_{\mathcal{S}_i} f(z)dz - \mu = \frac{1}{F_{\mu,\sigma}(v_{\min}, v_{\max})} \sum_{i=-k}^{0} (m_0 + iw) \int_{\mathcal{S}_i} f(z)dz - \mu$$

$$= \frac{1}{F_{\mu,\sigma}(v_{\min}, v_{\max})} m_0 \sum_{i=-k}^{0} \int_{\mathcal{S}_i} f(z)dz + \frac{1}{F_{\mu,\sigma}(v_{\min}, v_{\max})} w \sum_{i=-k}^{-1} i \int_{\mathcal{S}_i} f(z)dz - \mu$$

$$= -\delta + \frac{1}{F_{\mu,\sigma}(v_{\min}, v_{\max})} w \sum_{i=-k}^{-1} i \int_{\mathcal{S}_i} f(z)dz. \tag{25}$$

Then

$$\mathcal{E}_{\text{discretization}} = -\delta = \delta \cdot \left(\frac{F_{0,1}(-\beta h, \beta h)}{2Z} - 1 + o(1)\right), \tag{26}$$

Besides,

$$-kw\frac{F_{0,1}\left(-\beta k, 0\right)}{2Z} \leq \frac{1}{F_{\mu,\sigma}(v_{\min}, v_{\max})} w \sum_{i=-k}^{-1} i \int_{\mathcal{S}_i} f(z)dz \leq -w\frac{F_{0,1}\left(-\beta(k+1), -\beta\right)}{2Z}\Big|_{h=0}.$$

This is still equivalent to (24) when $h = 0$.

Combining case of $h \geq 1$ and case $h = 0$, we obtain the conclusion corresponding to case (i) in Theorem (3.1).

• **Case (ii)** $\mu \geq v_{\text{max}}$ .

$$
\begin{aligned}
\mathcal{E}_{a,b,m,\sigma} &= \sum_{j=1}^{m} z_i p_i - \mu \\
&= \frac{1}{F_{\mu,\sigma}(v_{\min}, v_{\max})} \sum_{i=-k}^{-h} m_i \int_{\mathcal{S}_i} f(z) dz - \mu = \frac{1}{F_{\mu,\sigma}(v_{\min}, v_{\max})} \sum_{i=-k}^{-h} (m_0 + iw) \int_{\mathcal{S}_i} f(z) dz - \mu \\
&= \frac{1}{F_{\mu,\sigma}(v_{\min}, v_{\max})} m_0 \sum_{i=-k}^{-h} \int_{\mathcal{S}_i} f(z) dz + \frac{1}{F_{\mu,\sigma}(v_{\min}, v_{\max})} w \sum_{i=-k}^{-h} i \int_{\mathcal{S}_i} f(z) dz - \mu \\
&= (m_0 - \mu) + \frac{1}{F_{\mu,\sigma}(v_{\min}, v_{\max})} w \sum_{i=-k}^{-h} i \int_{\mathcal{S}_i} f(z) dz \\
&= -\delta + \frac{1}{F_{\mu,\sigma}(v_{\min}, v_{\max})} w \sum_{i=-k}^{-h} i \int_{\mathcal{S}_i} f(z) dz.
\end{aligned}
\tag{27}
$$

Repeating the previous reasoning, we can obtain that

$$
\mathcal{E}_{\text{discretization}} = -\delta,
\tag{28}
$$

and

$$
-kw \frac{F_{0,1}(-\beta k, -\beta h)}{2Z} \leq \mathcal{E}_{\text{truncation}} \leq -(h+1)w \frac{F_{0,1}(-\beta(k+1), -\beta(h+1))}{2Z}.
\tag{29}
$$

This yields the conclusion corresponding to case (ii) in Theorem (3.1). $\qquad \square$

*Proof of Theorem 3.2.* Now we consider the two cases in Theorem 3.2 separately.

Case (i). If $\mu \in (v_{\min}, v_{\max})$ and $h \geq 1$, for the discretization error, the linear coefficient of $\delta$ is $\left( \frac{F_{0,1}(-\beta h, \beta h)}{2Z} - 1 + o(1) \right)$. Considering the fact that

$$
F_{0,1}(-\beta h, \beta h) = 1 - 2\Phi(-\beta h) = 1 - 2 \frac{1}{\beta h \sqrt{\pi}} e^{-\frac{(\beta h)^2}{2}} + o(1),
\tag{30}
$$

and

$$
\begin{aligned}
2Z = F_{0,1}(-\beta k, \beta h) &= 1 - \Phi(-\beta k) - \Phi(-\beta h) \\
&= 1 - \frac{1}{\beta h \sqrt{\pi}} e^{-\frac{(\beta h)^2}{2}} - \frac{1}{\beta k \sqrt{\pi}} e^{-\frac{(\beta k)^2}{2}} + o(1),
\end{aligned}
\tag{31}
$$

then

$$
\begin{aligned}
&\frac{F_{0,1}(-\beta h, \beta h)}{2Z} - 1 + o(1) \\
&\leq \left( 1 - 2 \frac{1}{\beta h \sqrt{\pi}} e^{-\frac{(\beta h)^2}{2}} + o(1) \right) \Big/ \left( 1 - \frac{1}{\beta h \sqrt{\pi}} e^{-\frac{(\beta h)^2}{2}} + o(1) \right) - 1 + o(1) \\
&= \frac{1}{\beta h \sqrt{\pi}} e^{-\frac{(\beta h)^2}{2}} + o(1) \\
&= C_{\beta,1} \cdot \frac{1}{h} e^{-h^2} + o(1),
\end{aligned}
\tag{32}
$$

(32) yields the linear coefficient of $\delta$ in case (i).

We further consider the truncation error,

$$
\begin{aligned}
\mathcal{E}_{truncation} &\leq kw\frac{F_{0,1}(-\beta k, -\beta h)}{2Z} \leq kw\Phi(-\beta h)/2 \approx kw\frac{1}{\beta h\sqrt{2\pi}}e^{-(\beta h)^2/2} \\
&\leq mw\frac{1}{\beta h\sqrt{2\pi}}e^{-(\beta h)^2/2} = C_{\beta,m,2}\cdot\frac{1}{h}e^{-h^2}\cdot w,
\end{aligned}
\tag{33}
$$

thus (33) yields the truncation error in case (i).

Case (ii). If $\mu \in (v_{\min}, v_{\max})$ and $h = 0$, or $\mu \notin (v_{\min}, v_{\max})$, we have $\mathcal{E}_{\text{discretization}} = -\delta$ directly. At this time

$$
\begin{aligned}
|\mathcal{E}_{\text{truncation}}| &\geq (h+1)w\frac{F_{0,1}\left(-\beta(k+1), -\beta(h+1)\right)}{2Z} \\
&\geq (h+1)w\frac{F_{0,1}\left(-\beta(k+1), -\beta\right)}{F_{0,1}\left(-\beta(k+1), \beta\right)} \\
&\geq (h+1)w\frac{F_{0,1}\left(-\infty, -\beta\right)}{F_{0,1}\left(-\infty, \beta\right)} \\
&= C_\beta(h+1)w,
\end{aligned}
\tag{34}
$$

where $C_\beta = \frac{F_{0,1}(-\infty,-\beta)}{F_{0,1}(-\infty,\beta)}$.

Combining case (i) and (ii) yields Theorem 3.2.

$\square$

## B.3    Error Propagation of HL-Gaussian

Approximation Policy Iteration (API) is a popular iterative paradigm to find an approximate solution to the optimal value function $Q^*$. SAC and TD3 can be regared within this framework. It starts from a policy $\pi_0$, and then approximately evaluates that policy $\pi_0$, i.e. it finds a $Q_0$ that satisfies $\mathcal{T}^{\pi_0}Q_0 \approx Q_0$. Afterwards, it performs a policy improvement step, which is to calculate the greedy policy with respect to (w.r.t.) the most recent action-value function, to get a new policy $\pi_1$. The policy iteration algorithm continues by approximately evaluating the newly obtained policy $\pi_1$ to get $Q_1$ and repeating the whole process again, generating a sequence of policies and their corresponding approximate action-value functions $Q_0 \to \pi_1 \to Q_1 \to \pi_2 \cdots$.

Similarly, we can build API for the setting that value functions are learned by HL-Gaussian. Specifically, it also starts from a policy $\pi_0$, and then approximately evaluates that policy $\pi_0$, i.e. it finds a categorical distribution $\hat{p}^{(0)}$ that satisfies the Bellman equation $\mathbb{E}_{p^{(0)}}[z] \approx \mathbb{E}_{\hat{p}^{(0)}}[z]$, where $p^{(0)}$ comes from projecting $\mathcal{T}^{\pi_0}\mathbb{E}_{\hat{p}^{(0)}}[z]$ on $m$ discrete locations through equation (6). Afterwards, it performs a policy improvement step, which is to calculate the greedy policy with respect to (w.r.t.) the most recent action-value function, to get a new policy $\pi_1$. The policy iteration algorithm continues by approximately evaluating the newly obtained policy $\pi_1$ to get $\hat{p}^{(1)}$ and repeating the whole process again, generating a sequence of policies and their corresponding approximate action-value functions $\hat{p}^{(0)} \to \pi_1 \to \hat{p}^{(1)} \to \pi_2 \cdots$.

**Theorem B.1** ((Error Propagation for AHL-Gaussian)). *Let $K$ be a positive integer and $\nu$ be some distribution on $\mathcal{S} \times \mathcal{A}$. Then, for any sequence of functions $\{\hat{p}^{(k)}\}(0 \leq k < K)$, the following inequalities hold with a high probability:*

$$
\|\mathbb{E}_{\hat{p}^*}[z] - Q^{\pi_K}\|_{2,\nu} \leq \frac{2\gamma}{(1-\gamma)^2}\left(C_{v_{\min},v_{\max},r_{\max},\mathcal{P},\delta,\nu}\cdot C_\nu^{1/2}\max_{0\leq k<K}\varepsilon_k + \gamma^{\frac{K}{2}-1}r_{\max}\right),
\tag{35}
$$

*where we use $Q^{\pi_K}$ to represent the true value function of $\pi_K$, $r_{\max}$ is an upper bound reward, $\rho$ is the initial distribution, $\frac{d(\cdot)}{d_\nu}$ represents the density ratio of two distributions. $C_{v_{\min},v_{\max},r_{\max},\mathcal{P},\delta,\nu}$ is a constant dependent on $v_{\min}, v_{\max}, r_{\max}, \mathcal{P}, \delta, \nu$,*

$$
C_{p,\nu} = (1-\gamma)^2\sum_{k\geq 1}k\gamma^{k-1}\sup_{\pi_1,\ldots,\pi_k}\left\|\frac{d(\rho P^{\pi_1}\cdots P^{\pi_k})}{d\nu}\right\|_\infty.
$$

*and*

$$\varepsilon_k = \mathbb{E}_{(s,a) \sim \nu} \left[ D_{KL}(\hat{p}^{(k)}(s,a), p^{(k)}(s,a)) + \mathcal{E}^{(k),2}_{v_{\min}, v_{\max}, m, \sigma} + \frac{1}{D(s,a)} \right], \quad (36)$$

*with $D(s,a)$ being the number of $(s,a)$ pairs in replay buffer.*

This result implies that the uniform-over-all iterations upper bound $\max_{0 \le k < K} \varepsilon_k$ is the quantity that determines the performance loss. Next, we analyze each term in $\varepsilon_k$:

- The last term represents the approximation error caused by using transitions from the replay buffer to approximate the true transition model of the MDP. This error is unavoidable but will asymptotically approach 0 as the replay buffer grows.
- The first term corresponds to the KL-divergence term that HL-Gaussian aims to optimize. As long as sufficient gradient descent steps are performed during the policy evaluation process for each $\pi_i$, this term will become sufficiently small.
- Thus, only the second term, the projection error, remains unaddressed by the existing HL-Gaussian method. Its presence will clearly have a negative impact on the final performance, so our AHL-Gaussian further optimize $\mathcal{E}^{(k),2}_{v_{\min}, v_{\max}, m, \sigma}$ to keep the whole $\max_{0 \le k < K} \varepsilon_k$ small enough.

Before we provide the proof outline for Theorem B.1, we cite a lemma for standard API as follows:

**Lemma B.4** ((Error Propagation for API Farahmand et al. (2010))). *Let $p \ge 1$ be a real and $K$ be a positive integer. Then, for any sequence of functions $\{Q^{(k)}\}(0 \le k < K)$, and their corresponding Bellman residuals $\varepsilon_k = Q_k - T^\pi Q_k$, the following inequalities hold:*

$$\|Q^* - Q^{\pi_K}\|_{p,\nu} \le \frac{2\gamma}{(1-\gamma)^2} \left( C^{1/p}_{p,\nu} \max_{0 \le k < K} \|\eta_k\|_{p,\nu} + \gamma^{\frac{K}{p}-1} r_{\max} \right),$$

*where $r_{\max}$ is an upper bound on the magnitude of the expected reward function, $\eta_k$ is the bellman error $Q_k(s,a) - \mathcal{T}^{\pi_k} Q_k(s,a)$ and*

$$C_{p,\nu} = (1-\gamma)^2 \sum_{m \ge 1} m \gamma^{m-1} \sup_{\pi_1, \ldots, \pi_m} \left\| \frac{d(\rho P^{\pi_1} \cdots P^{\pi_m})}{d\nu} \right\|_\infty.$$

*Proof Outline of Theorem B.1.* Now we analyze the Bellman error $\eta_k$ as follows:

$$\begin{aligned}
|\eta_k(s,a)| &= |Q_k(s,a) - \mathcal{T}^{\pi_k} Q_k(s,a)| \\
&= |Q_k(s,a) - \hat{\mathcal{T}}^{\pi_k} Q_k(s,a) + \hat{\mathcal{T}}^{\pi_k} Q_k(s,a) - \mathcal{T}^{\pi_k} Q_k(s,a)| \\
&\le |Q_k(s,a) - \hat{\mathcal{T}}^{\pi_k} Q_k(s,a)| + |\hat{\mathcal{T}}^{\pi_k} Q_k(s,a) - \mathcal{T}^{\pi_k} Q_k(s,a)| \quad (37)
\end{aligned}$$

From Proposition 3.1, the first term is upper bounded by

$$|Q_k(s,a) - \hat{\mathcal{T}}^{\pi_k} Q_k(s,a)|^2 \le 2 \max(v_{min}, v_{max})^2 D_{KL}\left(\hat{p}(s,a), p(s,a)\right) + \mathcal{E}^{(k),2}_{v_{\min}, v_{\max}, m, \sigma}, \quad (38)$$

where we use $(\hat{p}(s,a)$ and $p(s,a)$ to represent the categorical distribution $[\hat{p}_1(s,a), \cdots, \hat{p}_m(s,a)]$ and $[p_1(s,a), \cdots, p_m(s,a)]$, respectively.

The second term comes from using interaction transitions to approximate the true MDP and it is proved in Kumar et al. (2020) that with probability $1 - \delta$,

$$|\hat{\mathcal{T}}^{\pi_k} Q_k(s,a) - \mathcal{T}^{\pi_k} Q_k(s,a)| \le \frac{C_{\delta, r_{max}, \mathcal{P}}}{\sqrt{D(s,a)}}, \quad (39)$$

where $C_{\delta, r_{max}, \mathcal{P}}$ is a constant dependent on $\delta$, $r_{max}$ and the MDP $\mathcal{P}$, and $D(s,a)$ is the number of $s - a$ pairs in replay buffer.

Putting (38), (39) and Lemma B.3 togather we can obtain Theorem B.1.

$\square$

# C   Experimental Details

In this section, we will provide a detailed introduction to the experimental details of combining AHL-Gaussian with different algorithms. We integrated AHL-Gaussian with Q-learning method DQN (Mnih et al., 2015), SAC (Haarnoja et al., 2018) and TD3 (Fujimoto et al., 2018b), respectively.

**DQN with AHL-Gaussian.**   In order to assess the influence of AHL-Gaussian within the DQN framework, we have chosen the original DQN, the traditional distributional approach C51, and the standard HL-Gaussian as our benchmarks. Our testing is conducted using the Atari 2600 game environment. To guarantee an equitable evaluation, the implementations of all algorithms are grounded in the codebase provided by the repository at `https://github.com/DLR-RM/stable-baselines3`. Additionally, we have ensured that the shared hyperparameters across these algorithms remain uniform. The hyperparameters for our algorithm are detailed in Tables 1 and 2.

Table 1: Hyperparameters of DQN with AHL-Gaussian

|  | Name of Hyperparameter | Value |
| --- | --- | --- |
| AHL-Gaussian | number of bins $m$ | 51 |
|  | ratio $w_\xi/\sigma_\xi$ | 1.5 |
|  | initial $\xi$ | 3 |
|  | learning rate | 1e-3 |
|  | Interval Update Frequency | 1:1 |
| DQN | total timesteps | 1e+7 |
|  | buffer size | 100000 |
|  | learning rate | 1e-4 |
|  | batch size | 32 |
|  | $\gamma$ | 0.99 |
|  | exploration initial epsilon | 1.0 |
|  | exploration final epsilon | 0.01 |

Table 2: Architecture of Q Network

| Layer | Type | Input Dim | Output Dim | Kernel Size | Stride | Activation |
| --- | --- | --- | --- | --- | --- | --- |
| 1 | Conv2d | $observation\_dim$ | 32 | 8x8 | 4 | ReLU |
| 2 | Conv2d | 32 | 64 | 4x4 | 2 | ReLU |
| 3 | Conv2d | 64 | 64 | 3x3 | 1 | ReLU |
| 4 | Flatten | - | - | - | - | - |
| 5 | Linear | $flatten\_dim$ | $action\_dim$ | - | - | ReLU |

Note: the $observation\_dim$ and the $action\_dim$ are the observation and action space of certain Atari environment. For example, the observation and action space of the SpaceInvaders are Box(0, 255, (210, 160, 3), uint8) and Discrete(6), respectively.

**SAC with AHL-Gaussian.**   In order to assess the impact of incorporating AHL-Gaussian into SAC algorithm, we have chosen the standard SAC and a version of SAC with a fine-tuned HL-Gaussian as our comparisons. Our experiments are conducted within the Gym MuJoCo simulation environment. To guarantee an equitable comparison, the implementation of all the algorithms adheres to the same codebase, which can be found at `https://github.com/pranz24/pytorch-soft-actor-critic`. Additionally, we have ensured that the shared hyperparameters across these algorithms remain uniform. The hyperparameters specific to our approach are detailed in Tables 3.

**TD3 with AHL-Gaussian.**   To evaluate the impact of integrating AHL-Gaussian into the TD3 algorithm, we have also chosen the vanilla TD3 and a version of TD3 enhanced with a finetuned HL-Gaussian as our comparing approaches. Our evaluation is also conducted within the Gym MuJoCo environment. In terms of a balanced comparison, we have implemented all algorithms using the codebase provided by TD3's original author, which can be found at `https://github.com/`

Table 3: Hyperparameters of SAC with AHL-Gaussian

|  | Name of Hyperparameter | Value |
|---|---|---|
| AHL-Gaussian | the number of bins $m$ | 51 |
|  | ratio $w_\xi/\sigma_\xi$ $\alpha$ | 1.5 |
|  | initial $\xi$ | 10 |
|  | learning rate | 1e-3 |
|  | Interval Update Frequency | 1:1 |
| SAC | total timesteps | 5e+6 |
|  | buffer size | 1000000 |
|  | learning rate | 3e-4 |
|  | batch size | 256 |
|  | $\gamma$ | 0.99 |
|  | target update interval | 1000 |
|  | automatic entropy tuning | True |
|  | number of hidden layers in critic | 2 |
|  | hidden dim of critic | 256 |
|  | number of hidden layers in actor | 2 |
|  | hidden dim of actor | 256 |

`sfujim/TD3`. We have also ensured uniformity in the hyperparameters across these algorithms. The hyperparameters specific to our approach are delineated in Tables 4.

Table 4: Hyperparameters of TD3 with AHL-Gaussian

|  | Name of Hyperparameter | Value |
|---|---|---|
| AHL-Gaussian | the number of bins $m$ | 51 |
|  | ratio $w_\xi/\sigma_\xi$ $\alpha$ | 1.5 |
|  | initial $\xi$ | 10 |
|  | learning rate | 1e-3 |
|  | Interval Update Frequency | 1:1 |
| TD3 | total timesteps | 5e+6 |
|  | buffer size | 1000000 |
|  | learning rate | 3e-4 |
|  | batch size | 256 |
|  | $\gamma$ | 0.99 |
|  | std of Gaussian exploration noise | 0.1 |
|  | target network update rate | 0.005 |
|  | noise added to target policy during critic update | 0.2 |
|  | range to clip target policy noise | 0.5 |
|  | frequency of delayed policy updates | 2 |
|  | number of hidden layers in critic | 2 |
|  | hidden dim of critic | 256 |
|  | number of hidden layers in actor | 2 |
|  | hidden dim of actor | 256 |

**Interval Fine-tuning for HL-Gaussian.** To ensure that the baseline HL-Gaussian performs optimally and to conduct a more equitable comparison, we've adopted a fine-tuning strategy. This is because an ill-considered range for $[v_{\min}, v_{\max}]$ can significantly impair the effectiveness of HL-Gaussian. Our approach begins with running the original algorithms, SAC and TD3, on a designated task to ascertain the value function's settled value, $v_{\text{final}}$. Subsequently, we experiment with HL-Gaussian using $\xi$ values from the potential candidates: $[0.5v_{\text{final}}, 0.75v_{\text{final}}, v_{\text{final}}, 1.5v_{\text{final}}, 2v_{\text{final}}]$. We then identify the most effective $\xi$ from this selection to utilize as our chosen parameter.

# D Supplementary Experimental Results

Complete results for the ablation study are shown in Figure 11-Figrue 13.

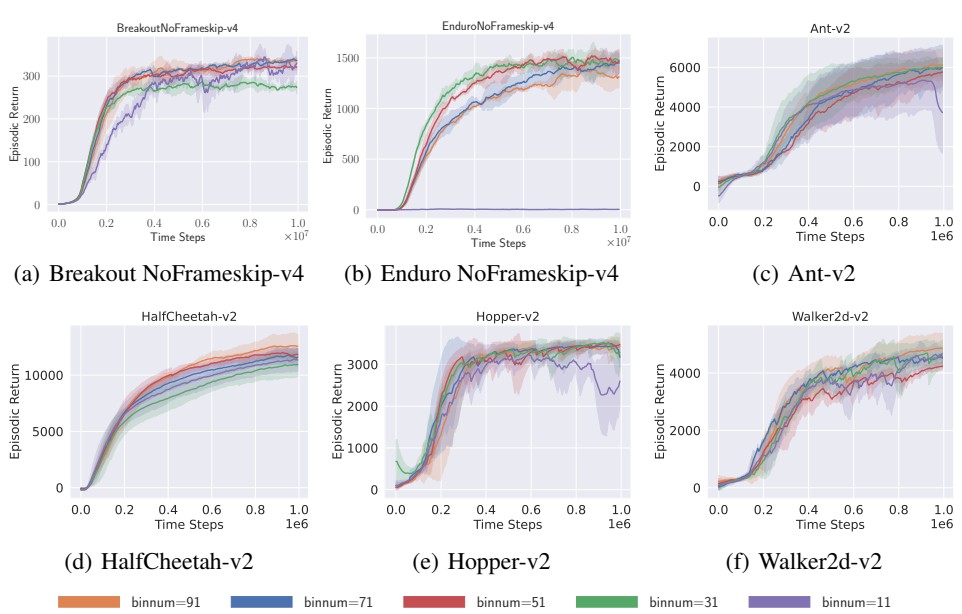

Figure 11: Ablation for the number of bins $m$.

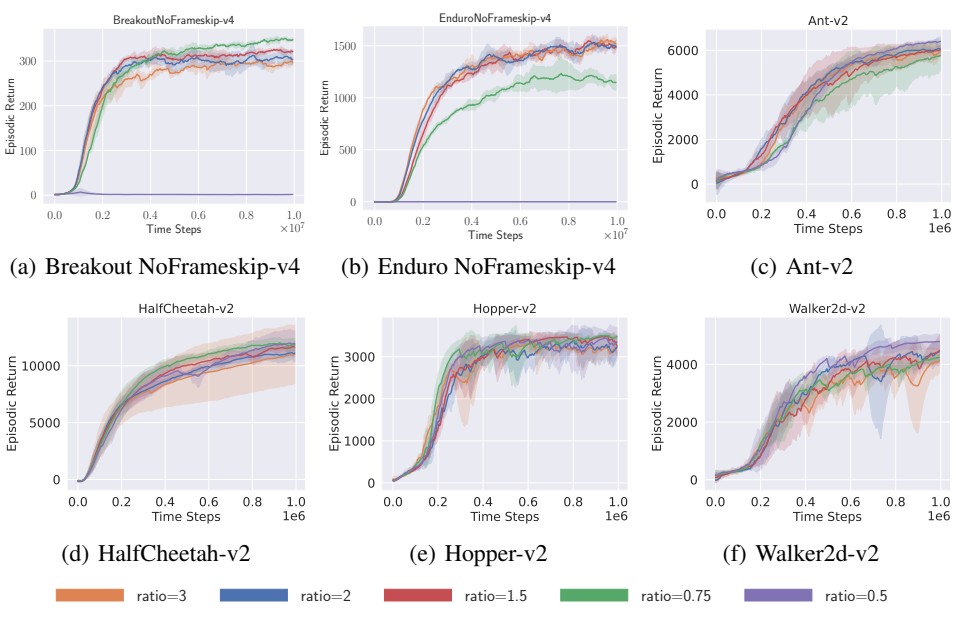

Figure 12: Ablation for the ratio of width to variance

# E Training Curves of $\xi$ and Projection Error

To further analyze the learning process of AHL-Gaussian, we plot the support interval bound $\xi$ and the projection error (as defined in equation 10) corresponding to the main performance curves

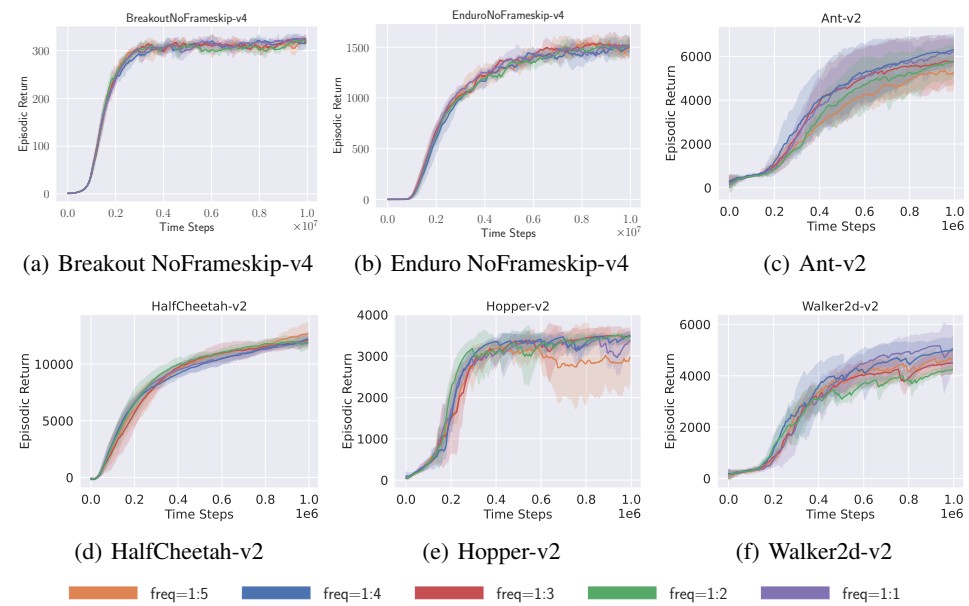

Figure 13: Ablation for the interval update frequency.

presented in Section 4.2. Specifically, Figures 14, 15, and 16 illustrate the results for DQN (Figure 4), SAC (Figure 5), and TD3 (Figure 6), respectively.

It can be observed that, in almost all tasks, the projection error exhibits a steep increase during the early stages. This is primarily due to the small initial interval range and the mismatch with the rapid growth of the value function in the early stages. As training progresses, the projection error gradually decreases to a sufficiently small value, indicating that the support interval $\xi$ effectively encompasses the value function.

Regarding the support interval bound $\xi$, it converges to a fixed value in most Atari tasks. However, in some Mujoco tasks, $\xi$ continues to show an upward trend even at 5M steps, particularly in HumanoidStandup and Humanoid. Interestingly, the scores for these two tasks also exhibit an upward trend, suggesting that as training progresses, the overall performance is likely to continue improving.

# F    Comparison with Other Non-learning Approaches

In this section, we incorporate three non-learning methods to adaptively adjust the support interval and compare them with our AHL-Gaussian. This is particularly relevant in MuJoCo, where each task has distinct reward magnitudes and ranges.

**Method 1**: The interval bound $\xi$ is set as the maximum value of all current Bellman targets, multiplied by a larger coefficient $\eta = 2$, as discussed in Section 4.3.

From Figure 17, it is evident that this approach performs comparably to AHL-Gaussian on certain tasks. However, it exhibits significant shortcomings in tasks such as Swimmer and HumanoidStandup. Specifically, the target values in the Swimmer task fluctuate drastically, making this "overly sensitive" adjustment method unable to converge to a reasonable interval, which severely impacts training performance.

**Method 2**: The support interval is set to $\left[\frac{r_{\min}}{1-\gamma}, \frac{r_{\max}}{1-\gamma}\right]$, where $r_{\min}$ and $r_{\max}$ are observed during training.

As shown in Figure 18, this method also demonstrates significant disadvantages in tasks such as Swimmer and HumanoidStandup. This is because it relies on discovering effective rewards during

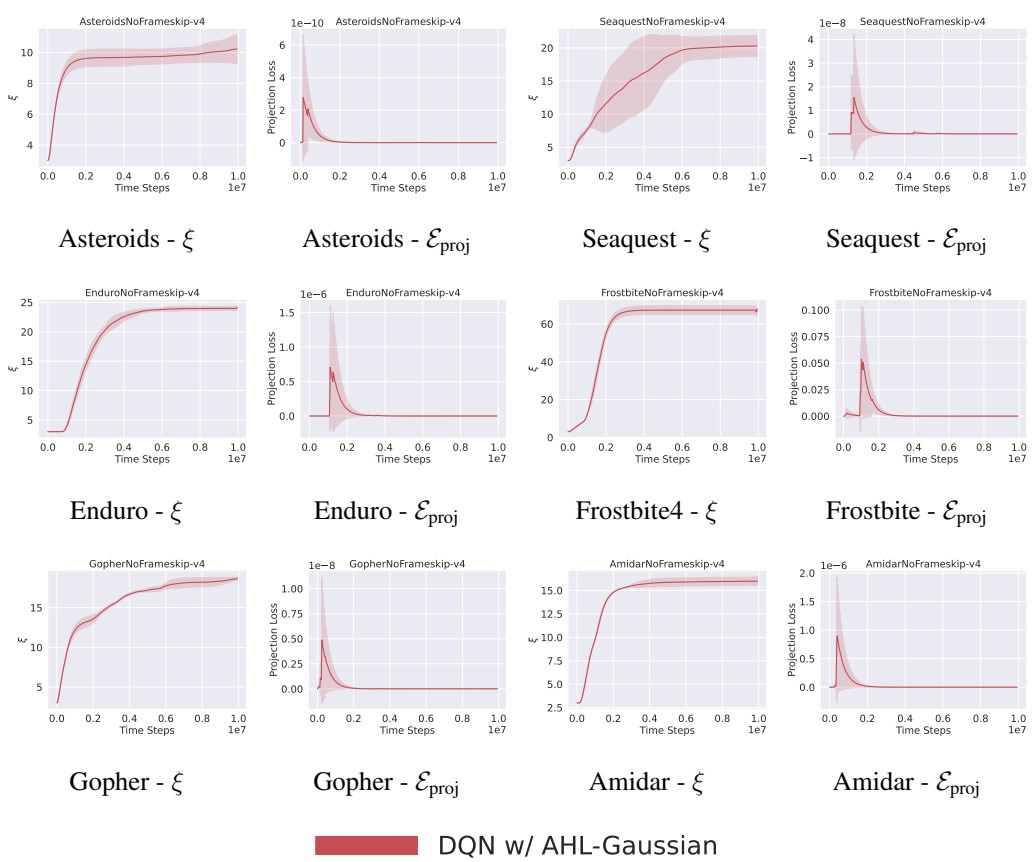

Figure 14: Projection error $\mathcal{E}_{\text{proj}}$ and $\xi$ in DQN.

training, rendering it unsuitable for tasks with highly fluctuating reward signals or those that are challenging to explore.

**Method 3**: This method utilizes prior knowledge of each task by normalizing rewards as $\frac{r}{r_{\max}}$, resulting in values within the range $[-1, 1]$. Consequently, the Bellman target is constrained to $[\frac{-1}{1-\gamma}, \frac{1}{1-\gamma}]$. With this approach, the support interval is fixed at $[-\frac{1}{1-\gamma}, \frac{1}{1-\gamma}]$.

Figure 19 shows that this method performs significantly worse than AHL-Gaussian across multiple tasks. This is likely because directly setting the support interval to $1/(1-\gamma)$ causes unnecessarily large projection errors during the early stages of training, when the received reward values are relatively small compared to the interval range. This negatively affects training performance. Moreover, obtaining a reliable prior estimate of the maximum reward is challenging in practice.

In summary, while these non-learning-based approaches achieve good performance on certain tasks, they have inherent limitations that make them unsuitable for a wide range of tasks. In contrast, our approach offers a more general and formal framework.

# G    Results on DM-Control

We also combined AHL-Gaussian with TD3 and conducted experiments on the more complex and sophisticated control tasks, Finger-Spin and Fish-Swim, in the DM Control suite, shown in Figure 20. The experimental results again demonstrate the advantages of AHL-Gaussian.

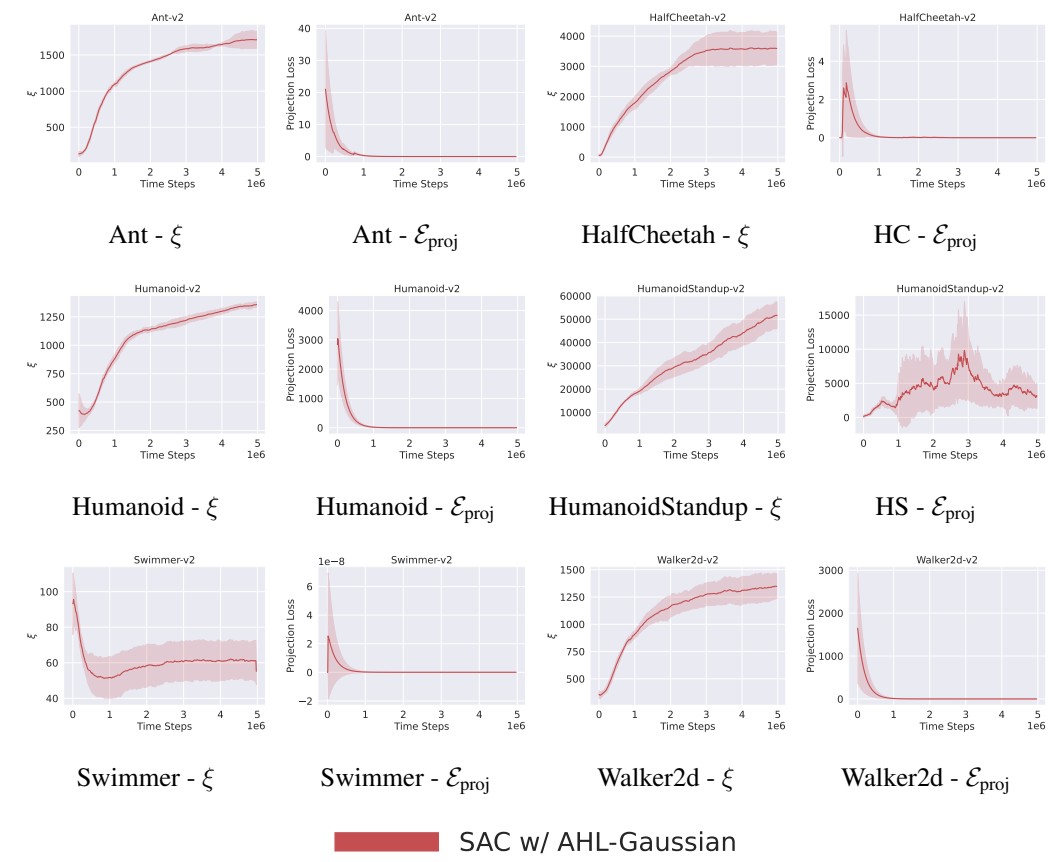

Figure 15: Projection error $\mathcal{E}_{\text{proj}}$ and $\xi$ in SAC.

# H    Atari Results with Longer Horizon

On the Atari tasks, we further extended the training horizon to observe the algorithm's performance. As shown in Figure 21, AHL-Gaussian significantly outperforms C51 across these three tasks. Additionally, Figure 22 illustrates the training curves of $\xi$ in AHL-Gaussian, revealing that $\xi$ steadily increases and converges to task-specific values.

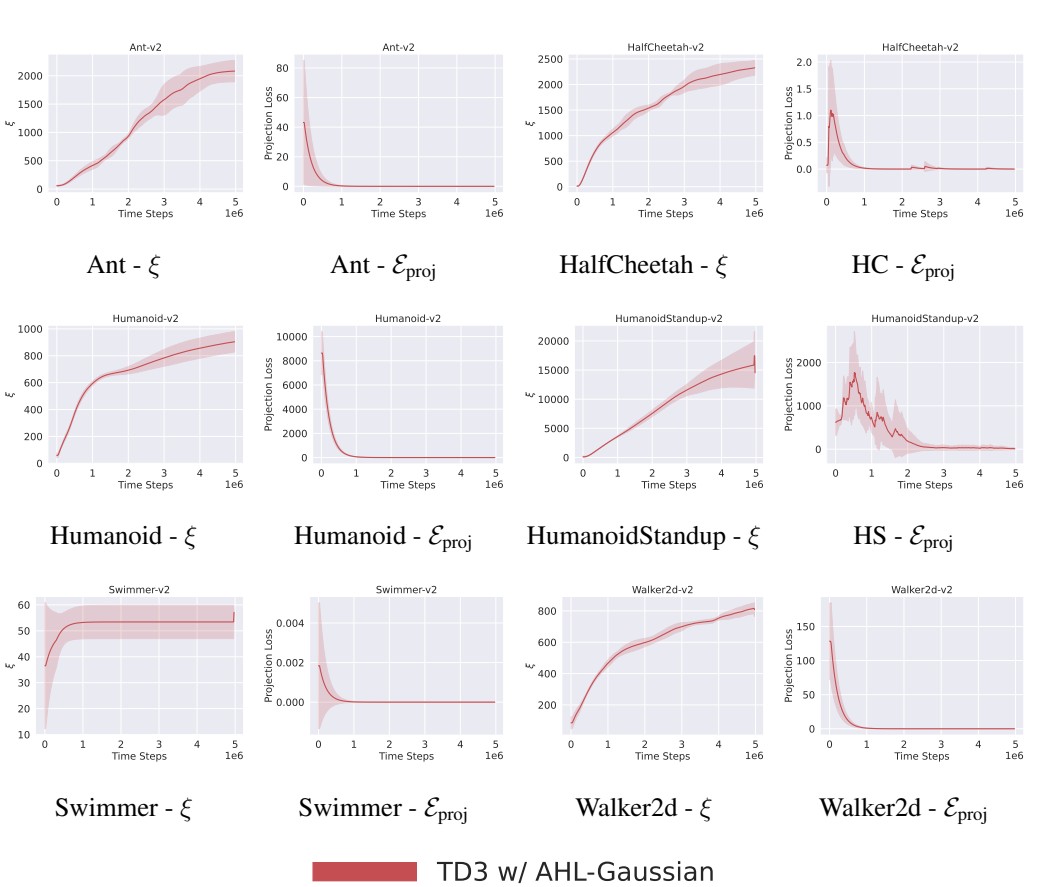

Figure 16: Projection error $\mathcal{E}_{\text{proj}}$ and $\xi$ in TD3.

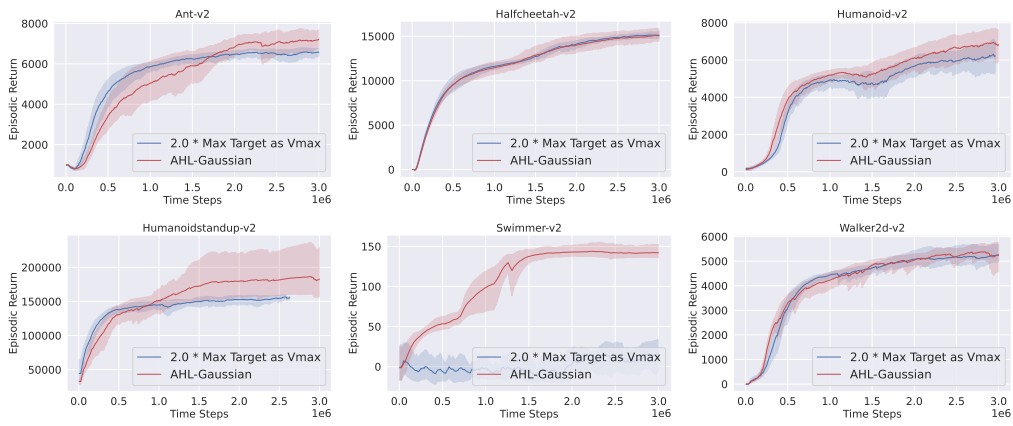

Figure 17: Performance Comparison between AHL-Gaussian and the non-learning-based method 1, where the interval bound $\xi$ is the maximum value of all current Bellman targets, multiplied by 2.

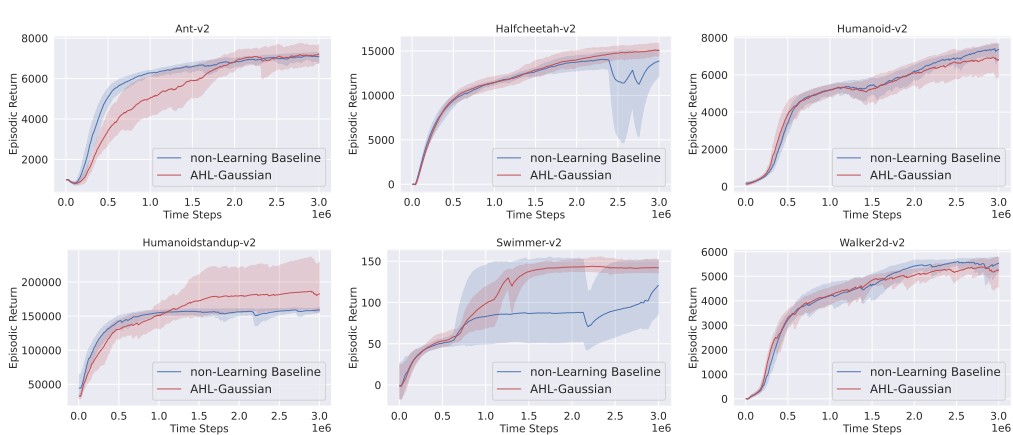

Figure 18: Performance Comparison between AHL-Gaussian and the non-learning-based method 2, where we set the support interval to $\left[\frac{r_{\min}}{1-\gamma}, \frac{r_{\max}}{1-\gamma}\right]$, where $r_{\min}, r_{\max}$ are observed during training.

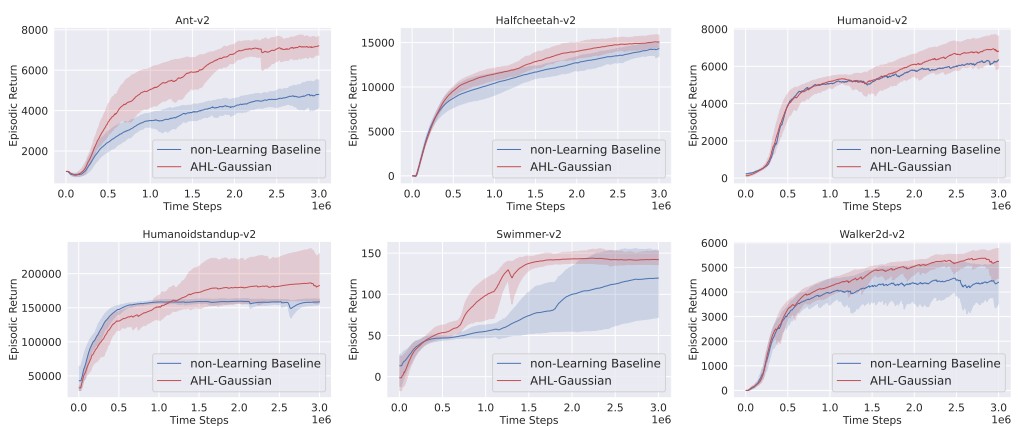

Figure 19: Performance Comparison between AHL-Gaussian and the non-learning-based method 3, where the reward is normalized and the support interval is fixed at $[-100, 100]$.

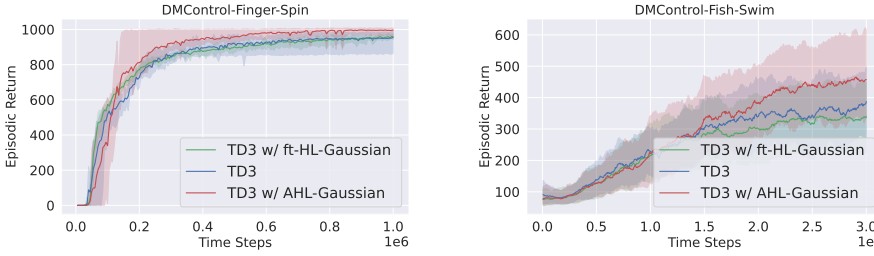

Figure 20: TD3 with AHL-Gaussian on DM Control Suite

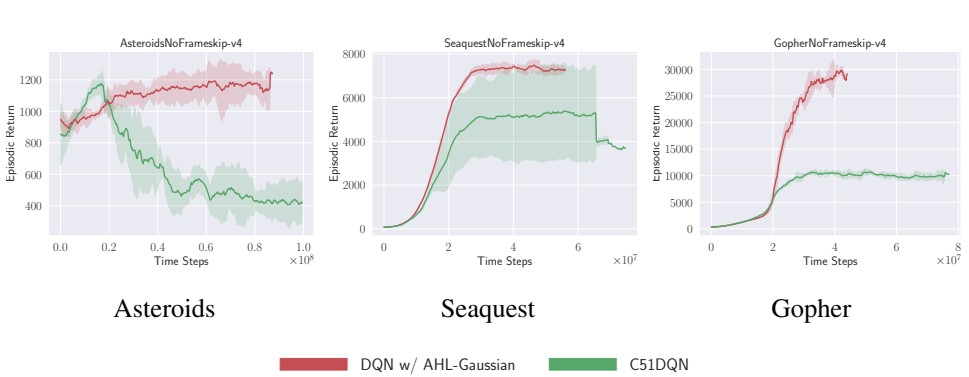

Figure 21: performance of AHL-Gaussian and C51 with a longer training horizon.

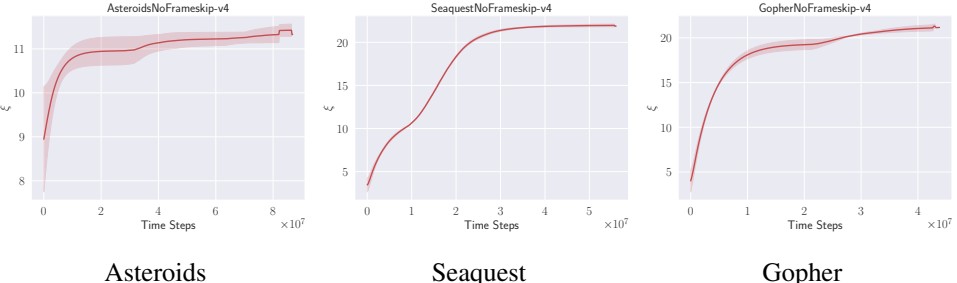

Figure 22: Training curves of $\xi$ with a longer training horizon.

