# OpenReview forum: "Adaptive HL-Gaussian: A Value Function Learning Method with Dynamic Support Adjustment"
_ICLR.cc/2025/Conference — Submitted to ICLR 2025_

### Official Review · Reviewer_GZrK · 2024-10-17

**Soundness:** 3
**Presentation:** 2
**Contribution:** 2
**Rating:** 5
**Confidence:** 4

**Summary:**

This paper proposed to strengthen the HL-Gaussian method by having dynamic interval on which the value function is projected into a categorical distribution. The existing works use a pre-specified fixed-length interval $[v_\text{min}, v_\text{max}]$ which can be hard to estimate. The authors conducted theoretical analysis and revealed that this projection incurs two losses:  projection loss and truncation loss. Based on the analysis, the authors proposed a learnable dynamic interval scheme in an attempt to minimize truncation error and projection error. The method was plugged into DQN and SAC and tested on standard benchmark environments.

**Strengths:**

The paper is overall well-written and sufficiently motivated. The preliminary section introduced the background and related work in an accessible way, paving the way for later analysis and proposal. Theoretical analyses are novel and intuitive, indicating an adaptive interval could provide an edge in minimizing the truncation and projection error. The proposed learnable scheme was combined with popular base algorithms to show quantitative improvements.

**Weaknesses:**

My concern for the paper is that its scope is limited and might not be interesting to the community, considering that the adaptive HL-Gaussian is an incremental improvement on the HL-Gaussian, and no significant higher scores can be seen from Figure 4-6. Maybe because the learning horizon was not suffiicently long, considering the original C51 algorithm ran 200m frames.

Another concern is the results seem to be limited as well. Central to the proposal, it would be more convincing to show how the dynamic interval $\zeta$ would change as learning proceeds, e.g. $\zeta$ can shrink drastically in some cases but being wide in the other. Consider those environments in figure 5 and 6 where no substantial difference was observed, was it because of $\zeta$ converged/flucturating around a number? And how was the loss of learning $\zeta$?  A question related to this is: if $\zeta$ did not change drastically but rather stayed around a fixed value, could we simply normalize the reward into $[0, 1]$, and replace the interval $[v_\text{min}, v_\text{max}]$ with $[0, \frac{1}{1-\gamma}]$? This reward normalization trick is frequently used and we do not need a separate learning scheme then.

**Questions:**

Please refer to the above for my questions.

---

> ### Author Response · Authors · 2024-11-16
> **Response to Reviewer GZrK**
>
> Thank you very much for your questions and suggestions. We will respond to each of them as follows:
>
> **1、About the significance and importance of the topic**
>
> First, although AHL-Gaussian is an incremental improvement over HL-Gaussian, HL-Gaussian has been shown to exhibit scaling laws, making it a highly promising foundational algorithm. Secondly, as the reviewer correctly pointed out, AHL-Gaussian has the potential to continue improving with more training steps. For example, in the Atari tasks, AHL-Gaussian still shows an upward trend at 5M steps. In Gym Mujoco, AHL-Gaussian also maintains an upward trend in tasks like Humanoid and HumanoidStandup.
>
> Moreover, we believe that this approach is not only applicable to the specific method of HL-Gaussian but also holds potential for extension to other Distributional RL methods that require a predefined support interval, such as C51 or ORDQN.
>
> **2、About the dynamic process of the interval bound  $\xi$ and the projection error**
>
> We would like to thank the reviewer for your suggestions. In the revision, we have added Section E to present the curves of $\xi$ and the projection error.
>
>
> It can be observed that, in almost all tasks, the projection error exhibits a steep increase during the early stages. This is primarily due to the small initial  interval range and the mismatch with the rapid growth of the value function in the early stages. As training progresses, the projection error gradually decreases to a sufficiently small value, indicating that the support interval $\xi$ effectively encompasses the value function. This phenomenon aligns with our theoretical predictions.
>
> Regarding the support interval bound $\xi$, it converges to a fixed value in most Atari tasks.
> However, in some Mujoco tasks, $\xi$ continues to show an upward trend even at 5M steps, particularly in HumanoidStandup and Humanoid. Interestingly, the scores for these two tasks also exhibit an upward trend, suggesting that as training progresses, the overall performance is likely to continue improving.
>
> **3、normalized the reward and use $[0,1/1-\gamma]$**
>
> We have added this non-learning baselines for further comparison in Section F. Figure  19 shows that this method performs significantly worse than AHL-Gaussian across multiple tasks. This is likely because directly setting the support interval to $1/(1-\gamma)$ causes unnecessarily large projection errors during the early stages of training, when the received reward values are relatively small compared to the interval range. This negatively affects training performance. Moreover, obtaining a reliable prior estimate of the maximum reward is challenging in practice.

---

> > ### Comment · Reviewer_GZrK · 2024-11-18
> >
> > Hi,
> >
> > Thanks for the clarification. But the revised paper has exceeded 10 pages which is not allowed.
> >
> > I appreciate the authors' efforts to include extra experiments to visualize $\zeta$ and such. My only concern is that the empirical results do not really well support superiority of the proposed dynamic $\zeta$. Even though AHL-Gaussian shows an upward trend at 5M steps, no clear evidence can be provided at such short horizon. It could hold the potential to be competitive to these baselines like C51, but running only $\frac{1}{20}$ of their horizon limits what we can conclude.

---

> > > ### Author Response · Authors · 2024-11-18
> > > **Response to Reviewer GZrK**
> > >
> > > Thank you very much for your feedback and thoughtful suggestions. We have adjusted the formatting of the main text to meet the page limit requirements.
> > >
> > > Additionally, in the current version, we conducted training for  5M steps on Gym Mujoco tasks.  5M is already the standard training horizon for most continuous action space algorithms and our method indeed demonstrates advantages over the baselines within 5M.
> > >
> > > As for the tasks on Atari, we have conducted experiments with a longer horizon. We will strive to update the results before the rebuttal deadline.

---

> > > ### Author Response · Authors · 2024-11-25
> > > **Experiments with a longer training horizon**
> > >
> > > Thanks a lot for your waiting and we sincerely apologize for the delay.
> > >
> > > Based on your suggestions, we have added Figure 21 in the revision  to include the performance of AHL-Gaussian and C51 under a longer training horizon. Due to computational and time constraints, the experiments could not reach 200M steps, and the progress varies slightly across tasks.
> > >
> > > **However, the results already demonstrate clear convergence, validating that AHL-Gaussian significantly outperforms C51 on these tasks**.
> > >
> > > Additionally, we present the training curve of $\xi$ for AHL-Gaussian in Figure 22, showing that $\xi$  increases monotonically and gradually stabilizes at a task-specific value, aligning perfectly with our theoretical predictions. Once the experiments are fully completed, we will update the results accordingly.
> > >
> > > We would greatly appreciate it if you could reevaluate the contributions of our work based on the latest results.

---

### Official Review · Reviewer_uVAy · 2024-11-02

**Soundness:** 1
**Presentation:** 1
**Contribution:** 2
**Rating:** 3
**Confidence:** 3

**Summary:**

The paper introduces a task-agnostic, value-function-aware mechanism for adaptively adjusting the intervals of HL-Gaussian, an existing approach that replaces the mean squared error loss function with cross-entropy loss in reinforcement learning. To achieve this, the paper provides a theoretical analysis of the errors arising from discretization and truncation in HL-Gaussian, guiding the development of a solution to minimize these errors. Experiments on Atari 2600 games and Gym Mujoco demonstrate the advantages of the proposed approach.

**Strengths:**

The paper addresses a novel problem: adaptively setting intervals for HL-Gaussian, distinguishing it from existing approaches that rely on fixed, predetermined intervals. This adaptive approach is motivated by a theoretical analysis of the discretization and truncation errors in HL-Gaussian.

**Weaknesses:**

The paper's major weaknesses include unclear notation and an insufficient connection between the theoretical analysis and the proposed solution.

Regarding notation issues, it may be partly due to my background, but the paper's notations are quite confusing, making it challenging to verify the accuracy of the theoretical analysis. Some specific examples include:

1. Equation (1) says $Q\^\*(s\_t,a\_t) = (T\^\* Q\^\*)(s\_t,a\_t)$. It seems to be incorrect as it means $T\^*$ is an identity function. The same issue is in Equation (2).

2. In lines 129–130, it appears that $z$ is treated as a random variable, yet the paper describes it as a distribution. Additionally, this distribution is said to be taken from a set of discrete locations, which seems unusual.

3. In Equation (3), $\delta_{z_i}$ is undefined. If it is the Dirac delta function, then is $Z(s_t,a_t)$ a probability? But if $Z(s_t,a_t)$ a probability, then then $Q(s_t,a_t)$ in Equation (3) is clearly incorrect.

4. In lines 138-139, why does $p_i(s_t,a_t)$ exist if the true Q value function does not belong to the interval $[v_{min}, v_{\max}]$?

5. Why does $p_i(s_t,a_t)$ in Equation (6) satisfy the condition in line 139?

6. In the proof of Proposition 3.1: What is "the distributional value function Q"? Why does $h(x)$ is supported in the range $[v_{min}, v_{max}]$? Why is $p_\mu$ independent of the learned variable $x:=(s,a)$ even though $p_{\mu} = [p_1(s,a),\dots,p_m(s,a)]$? There are also errors in lines 704, 711.

7. In line 198, $\delta$ is defined when there is a bin containing $\mu$. However, in line 212, it is used even for the case when $\mu$ is not in the interval (no bin containing $\mu$).

8. In line 212, $\mathcal{E}_{discretization} \le 0$, what does this mean?

9. In the theoretical analysis, the paper uses the same notation $Q$ and $\hat{T}$ for both the action-value functions computed using a continuous range and those computed using a truncated discrete range. However, these two procedures are different, and the error between them accumulates over the Markov chain in the Bellman equation. It appears that the paper does not address this error accumulation.

10. Line 777 is not acceptable in stating that a certain term is constant and that its value is $o(1)$ by "empirically verified". Additionally, the meaning of $o(1)$ in this context is unclear.

Regarding the solution, it appears to be largely disconnected from the theoretical analysis. Specifically, there is no assessment of the algorithm’s projection error. Furthermore, while this error depends on both the number of bins and the interval size, the proposed solution optimizes only the interval size while keeping the number of bins fixed. More importantly, the solution does not incorporate any findings from the theoretical analysis.

**Questions:**

Please see the above weaknesses.

---

> ### Author Response · Authors · 2024-11-16
> **Response to Reviewer uVAy**
>
> Thank you very much for your questions and suggestions. We will respond to each of them as follows:
>
> **About the unclear notations**
>
> We apologize for any confusion the reviewer may have had regarding some of the notations in the paper. We will clarify each of them below.
>
> **About 1:**   $\mathcal{T}^*$ and $\mathcal T^\pi$ are defined here as Bellman operators. . $\mathcal T^{*} Q(s_t,a_t)$ and $\mathcal T^\pi Q(s_t,a_t)$ represent the Bellman backups of $Q(s_t,a_t)$, and equations (1) and (2) represent the Bellman equations. This is the standard definition in  RL and is widely used in RL literature.
>
> **About 2 and 3:** In this paper, $Q(s, a)$ is represented as the expected value of a random variable $Z(s,a)$, where $Z(s,a)$ follows a categorical distribution . The discrete values of this categorical distribution consist of $m$ locations $[z_1, \cdots, z_m]$, each lying within the interval $[v_{\text{min}}, v_{\text{max}}]$. We have emphasized in the main text that $Z$ is a random variable, not a distribution, and provided the definition of $\delta(\cdot)$, highlighted in blue. We believe the revised explanation is more rigorous and easier to understand.
>
> **About 4 and 5:**  We first introduce the basic assumption of Distributional RL methods: the random variable $Z(s,a)$ corresponding to $Q(s,a)$ lies within a bounded region $[v_{\text{min}}, v_{\text{max}}]$ and follows a categorical distribution $[\hat p_1(s,a), \cdots, \hat p_m(s,a)]$. At the same time, the Bellman target $\mathcal{T}Q(s,a)$ is assumed to be the expectation of some random variable $Z'(s,a)$, where $Z'(s,a)$ follows a categorical distribution $[p_1(s,a), \cdots, p_m(s,a)]$, which also explains why the condition $\sum_{i=1}^{m} p_i(s,a) z_i = \mathcal{T}Q(s,a)$ needs to hold in equation (139).
>
> However, if the pre-specified $[v_{\text{min}}, v_{\text{max}}]$ is unreasonable, such that the true $Q$ values may fall outside this range, particularly when the Bellman target is more likely to fall outside $[v_{\text{min}}, v_{\text{max}}]$, the assumptions of the existing framework become invalid. In this case, the equation $\sum_{i=1}^{m} p_i(s,a) z_i = \mathcal{T}Q(s,a)$ may no longer hold.
>
> Therefore, we propose an adaptive adjustment of $[v_{\text{min}}, v_{\text{max}}]$ to ensure that the equation $\sum_{i=1}^{m} p_i(s,a) z_i = \mathcal{T}Q(s,a)$ holds with only minimal error, thereby better enabling the application of Distributional RL methods for learning.
>
> **About 6:**  We sincerely apologize for the typo in the proof of Proposition 3.1 and for the lack of further explanation regarding some of the symbols, which led to confusion for the reviewers. We have corrected the expression in this part of the proof in the revision and highlighted it in blue. In fact, we introduced the notation $[p_1(s,a), \cdots, p_m(s,a)]$ as $p_{\mu}$ and $[\hat p_1(s,a), \cdots, \hat p_m(s,a)]$ as $h_x$ simply to apply Lemma B.1. Additionally, the assumption that $h_x$ has supports in $[v_{\text{min}}, v_{\text{max}}]$ is a fundamental assumption for these methods, as described in the previous response.
>
> **About 7:**  Thank you very much for pointing out the issue with the imprecise definition of $\delta$. In fact, even when $\mu$ does not belong to $[v_{\text{min}}, v_{\text{max}}]$, we can still assume the existence of bins, allowing us to define $m_0$ and $\delta$. This does not affect the subsequent proof. We have added a footnote in the main text to clarify this point.
>
> **About 8:**   Since $\delta$ represents the distance between $\mu$ and the center of a bin, $m_0$, it can be either positive or negative. Therefore, $\mathcal{E}_{discretization}$ can also be both positive and negative, which means it can be less than 0.
>
> **About 10:** We have removed the inappropriate expression in Line 777.  o(1) here represents a constant much smaller than 1, and we have provided its definition in Line 215.

---

> ### Author Response · Authors · 2024-11-16
> **Response to Reviewer uVAy(2)**
>
> **About 9:**
>
>  We will provide a detailed comparison of the Q-function learning process under the traditional framework and the distributional RL framework to align our understanding:
>
> 1) In the traditional Q-learning framework, $Q_{\text{trad}}(s, a) $itself is the variable to be learned and iterated. In the distributional RL framework discussed in this paper, $Q_{\text{distri}}(s, a) = \sum_{i=1}^{m} \hat p_i(s, a) z_i $, where $\hat p_i(s, a) $ is the variable to be learned and iterated. Although the forms of the Q-functions differ slightly, $ Q_{\text{distri}} $  can be understood as a more complex representation of the Q-function, without any fundamental difference in meaning.
>
> 2) In the traditional Q-learning framework, $ Q_{\text{trad}}(s, a) $  is learned by fitting the Bellman equation through optimizing the MSE loss (or Bellman residual):
> $$
> \min_{Q_{\text{trad}}} \left[ Q_{\text{trad}}(s, a) - \hat{\mathcal T} Q_{\text{trad}}(s, a) \right]^2.
> $$
> In contrast, in the framework discussed in this paper,$ Q_{\text{distri}} $ is learned by optimizing the KL-divergence between $[\hat p_i(s, a)] \) and \( [p_i(s, a)] $. Here,
> $$
> \hat{\mathcal T} Q_{\text{distri}}(s, a) = r(s, a) + \gamma Q_{\text{distri}}(s', a'),
> $$
> where $ a' \sim \pi(s) \) or \( a' = \max_a Q_{\text{distri}}(s', a) $, and $ [p_i(s, a)] $ is obtained by projecting $\hat{\mathcal T} Q_{\text{distri}}(s, a) $ on the support interval. From this, it can be seen that although the two frameworks differ in the form of the Q-function, the ultimate goal remains the same: to ensure the Q-function satisfies the Bellman equation. Therefore, determining the performance of the policy evaluation  algorithms under the two frameworks only requires evaluating whether the Bellman equation is satisfied, i.e.,whether  the Bellman residual is small.
>
> 3) However, during the projection of $ \hat{\mathcal T} Q_{\text{distri}}(s, a) $ into $ [p_i(s, a)] $, an inappropriate choice of the interval may introduce extra projection errors, thereby increasing the Bellman residual and affect negatively the policy evaluation process.  The impact of the projection error has been elaborated in detail  in Proposition 3.1. This work thus aims to minimize the projection error by adjusting the interval.
>
> 4) To further demonstrate the influence of projection error in each policy evaluation step on the final learned Q-value, we have added Section B.3 in the appendix. These results aim to show the cumulative effects of projection error at each policy evaluation step and highlight the importance of optimizing the projection error during training.
>
> **About the relationship between solution and theory**
>
> Our theory conveys the following key points:
>
> **First**, Proposition 3.1 demonstrates that HL-Gaussian introduces projection error, which increases the Bellman residual and thus negatively affects policy evaluation, leading to the conclusion that reducing projection error is necessary.
>
>  **Additionally**, we have added Section B.3 in the appendix to discuss the cumulative effect of the projection error in one-step policy evaluation on the algorithm's final performance, highlighting the importance of minimizing projection error during each policy evaluation step.
>
> **Further**, Theorem 3.2 estimates the numerical range of the projection error and shows that the magnitude of this error depends on the positional relationship between the Bellman target $\mu$ and the interval $[v_{\text{min}}, v_{\text{max}}]$, summarized in the key finding highlighted in the gray box. This finds reveal that the projection error increases markedly if $[v_{\text{min}}, v_{\text{max}}]$ does not match the current target Q values.
>
> **Finally**, it is precisely because of our theoretical findings that we are inspired to treat $[v_{\text{min}}, v_{\text{max}}]$ as learnable variables and adaptively adjust them by optimizing projection error, forming the basis of our algorithm design.
>
> **We believe that our theoretical insights and algorithm design are logically consistent and clearly structured.**
>
> The reason we do not optimize the number of bins $m$is that this would lead to a time-varying network architecture, which poses implementation challenges. Our ablation study also shows that the algorithm's performance is robust when $m$ is within a certain range.
>
> Besides, we have also added Section E in the appendix to visualize the training curves of projection error  and the learned interval and their trends also align with our theoretical predictions.

---

> ### Author Response · Authors · 2024-11-25
> **Looking forward to Continuing the Discussion**
>
> Dear Reviewer uVAy:
>
> We sincerely thank you for your efforts and valuable feedback on reviewing our paper. We noticed that you have not yet participated in the discussion, and we would like to ask if you have any additional comments or questions that we can address collaboratively. We have revised the manuscript based on your insightful feedback and are eager to address any remaining concerns.
>
> Please let us know if you have any further thoughts or questions. We look forward to continuing our discussion.
>
> Best regards,
>
> Authors

---

> > ### Comment · Reviewer_uVAy · 2024-11-26
> >
> > I appreciate the authors' response and the revisions to the paper, which addressed some of my concerns. However, a few issues remain unresolved.
> >
> > 1. $Z$ is said to be a random variable. But in Equation (3) in the revised paper, $Z(s\_t, a\_t) = \sum\_{i=1}^m \hat{p}\_i(s\_t,a\_t) \delta\_{z\_i}$. Is this sum a random variable?
> > 2. $\mathcal{L}\_{MSE}$ in Equation (8) is expected to represent the traditional TD error, where the Q-value function should be calculated without any projection onto the support interval or discretization. However, lines 707-710 in the revision say that $Q(s,a) = E_{h_x}[z(x)]$ which means $Q(s,a)$ is projected onto the support interval and discretized. This means $\mathcal{L}_{MSE}$ is computed using the projected and discretized Q values, which is not the traditional TD error.
> > 3. The connection between the theory and the algorithm is unclear. For instance, it is not evident from the theory how the proposed algorithm reduces the projection error, nor does it seem to leverage certain theoretical insights, such as the positional relationship between the Bellman target and the interval. Overall, the theory primarily highlights that a fixed interval introduces projection error, while the algorithm addresses this by performing gradient descent to optimize the interval and minimize the error.

---

> ### Author Response · Authors · 2024-11-26
> **Response to Reviewer uVAy**
>
> Thank you very much for your response. We are pleased to know that our revised manuscript and replies have addressed some of your concerns. Below are our responses to the remaining concerns:
>
> **Question 1:** $Z(s_t,a_t)$ is a random variable that follows a categorical distribution, where the probability of taking the value $z_i$ is $\hat p_i(s_t, a_t)$. We followed the definition in paper [1] to write $Z(s_t,a_t)$ as $Z(s_t, a_t) = \sum_{i=1}^m \hat p_i(s_t, a_t) \cdot \delta_{z_i}$. However, we agree that this expression may cause misunderstanding, so we have removed it.
>
> **Question 2:** We suspect there might be a difference in how we define **traditional TD error**. We infer that, in your context, **traditional TD error** strictly requires the Q-value to be a scalar rather than the expectation of a random variable Z. However, in our context, the Q-value can be the expectation of a random variable, as long as the error takes the form $[Q(s,a)-\mathcal{T}Q(s,a)]^2$, rather than the cross-entropy induced by $p$ and $\hat p $. Thus, our Proposition 3.1 provides a comparison between these two error forms.
>
> From a different perspective, within HL-Gaussian, the Q-value can still be interpreted as a scalar rather than the expectation of a random variable. The distinction lies in how this scalar is computed: previously, Q value was directly the output of a neural network (NN), whereas now it is obtained by applying a softmax function to the output of the NN's last layer, followed by a dot product with $\{z_1, \cdots, z_m\}$. This modification introduces a more sophisticated NN structure. Consequently, we believe it is reasonable to express $[Q(s,a)-\mathcal T Q(s,a)]^2$ as **traditional TD error**, as it continues to represent the loss function that traditional RL methods aim to optimize.
>
> **Question 3:** We believe your understanding that "the theory primarily highlights that a fixed interval introduces projection error, while the algorithm addresses this by performing gradient descent to optimize the interval and minimize the error" is  correct. However, we would like to further highlight the intrinsic connection between the theory and the algorithm:
>
> 1. Our theory identifies the source of projection error and explains how it influences the projection error. Based on this insight, **we choose to optimize the interval range, which is the root cause, instead of selecting some other unrelated variables.**
>
> 2. Furthermore, our theory predicts that a reasonable interval range should precisely cover the current Q-values without being too large to cause unnecessary errors. This also explains **the necessity of dynamically adjusting the interval range as the Q-values evolve**.
>
> Thank you again for your feedback and suggestions. We sincerely hope this response addresses all your concerns.
>
> [1] Stop regressing: Training value functions via classification for scalable deep rl.

---

> > ### Comment · Reviewer_uVAy · 2024-11-27
> >
> > I would like to clarify that these remain my original concerns and are not new issues. Specifically:
> >
> > * The concern that $Z(s_t, a_t)$ is not a random variable as per its definition was raised in point 3.
> >
> > * In point 9, I expressed confusion about when $Q$ is defined using discretization versus when it uses continuous values. I reiterated this concern in the revised question 2 above.
> >
> > * Regarding question 3, it aligns with my initial concern that there is limited connection between the theory and the algorithm.
> >
> > I still find it strange that $Z(s_t, a_t) = \sum_{i=1}^m \hat p_i(s_t, a_t) \cdot \delta_{z_i}$ is considered a random variable (it looks like the expression of an expectation).
> >
> > Your inference is correct. I was thinking that the traditional TD error typically involves a scalar value of $Q$ values. When $Q$ values are restricted to the expectation of a categorical distribution (with the support [v_min, v_max]), there is a potential introduction of error. To analyze this projection error, one should compare the traditional TD error (mean squared error, MSE, using scalar $Q$ values) with the cross-entropy loss (where $Q$ values represent the expectations of a categorical random variable).
> >
> > From the author’s response, it seems the projection error discussed in the paper stems solely from the difference between MSE and cross-entropy loss, as both in Proposition 3.1 assume $Q$ values are expectations of categorical random variables. Is that correct?
> >
> > Furthermore, the argument that $Q$ values can still be interpreted as scalars through a more complex neural network may not hold. This is because the support [v_min, v_max] of the categorical random variables imposes constraints. The expectation can only represent any scalar value when there are no such restrictions on the distribution's support (or the distribution, in general).

---

> ### Author Response · Authors · 2024-11-27
> **Response to Reviewer uVAy**
>
> Thank you very much for your quick response. We greatly appreciate the discussion process  as it will certainly help make our work better understood and explained.
>
> 1. First, we apologize for the wording "new questions" in our previous reply. What we intended to express was the questions you raised in **the new reply**, **not additional questions**. We have already revised the wording in our previous reply.
>
> 2. The expression $\sum_{i=1}^m \hat p_i(s_t, a_t) \cdot \delta_{z_i}$ is meant to represent a categorical distribution, not an expectation. Therefore, a more accurate expression would be $Z(s_t, a_t) \sim \sum_{i=1}^m \hat p_i(s_t, a_t) \cdot \delta_{z_i}$, where $\sim$ replaces the equality sign $=$. We also agree that the previous equality was not accurate, and as such, we have removed it in the revision and replaced it with a description to reduce the potential for misunderstanding.
>
> 3. Your understanding of "the projection error discussed in the paper stems solely from the difference between MSE and cross-entropy loss, as both in Proposition 3.1 assume values are expectations of categorical random variables" is correct. However, we would like to explain more about TD-error and our insights.
>
>   * Firstly, we understand that the TD error should be induced by the algorithm and the underlying function space. When the algorithm restricts the function space used to express $Q$, the TD error should be the MSE loss between $Q$ and $\mathcal T Q $, where $Q$ is in the chosen function space. The error between the Q-value generated by the algorithm and the true $Q^{\pi} $ should be defined as the "approximation error." This is, of course, important but does not belong to the category of TD error.
>
>    * Secondly, in the widely used neural networks (NNs), additional constraints such as ReLU activation functions and layer normalization are commonplace. These constraints undoubtedly influence the output range of the neural network. Therefore, although HL-Gaussian restricts the Q-value range to $[v_{\min}, v_{\max}]$, it is essentially no different from the above-mentioned treatments.
>
>    * Finally, as you mentioned, constraining the Q-values to $[v_{\min}, v_{\max}]$ does indeed introduce error, i.e., "approximation error." Therefore, our approach adjusts $[v_{\min}, v_{\max}]$ to "reduce the error introduced by target projection," ensuring that the Q-value always stays within the range of $[v_{\min}, v_{\max}]$. This approach helps to mitigate the "approximation error." Although this point has not yet been explicitly stated in the theoretical expression, it is indeed a key insight in our theoretical work.
>
> We hope our reply helps clarify your concerns, and  please do not hesitate to let us know if you have any new comments.

---

> > ### Comment · Reviewer_uVAy · 2024-11-27
> >
> > Thank you for your prompt response. I appreciate your explanation. That said, I believe that if Proposition 3.1 assumes Q-values to be the expectations of a categorical random variable within the support [v_min,v_max], the connection between the theory and the algorithm might become less clear. This is because the error analysis seems to focus on the difference between the cross-entropy loss and the MSE, rather than addressing the error introduced by the restriction to the support [v_min,v_max].

---

> > > ### Author Response · Authors · 2024-11-27
> > > **Response to Reviewer uVAy**
> > >
> > > Thank you very much for your quick feedback. We are also pleased that our previous response addressed your concerns. However, we would like to further elaborate on some aspects you may have overlooked.
> > >
> > > First, according to Lemma B.4 in our revision, the TD-error is the key factor determining the algorithm's suboptimality gap (i.e., $(Q^{\pi_K}-Q^*)^2$), where \(K\) represents the number of policy iterations. Therefore, as long as the TD-error $(Q-\mathcal{T}^{\pi_k} Q)^2$ is sufficiently small for each intermediate policy $\pi_k, k \leq K$, the algorithm can ensure a small suboptimality gap. This implies that analyzing the error between Q and the true $Q^{\pi_k}$ is not strictly necessary.
> > >
> > > Moreover, Proposition 3.1 reveals that the upper bound of the TD-error is related to the cross-entropy loss and the projection error. This insight motivates the design of our algorithm, which simultaneously optimizes both the cross-entropy and the projection error, so as to obtain the final small suboptimality gap. Therefore, we believe the theory and the algorithm are closely connected.
> > >
> > > Under the mechanism of our algorithm, it effectively mitigates the approximation error introduced by restricting Q to $[v_{\min}, v_{\max}]$. At the early stage of training, both the Q-values and the true $Q^{\pi_k}$ values are relatively small and fall within $[v_{\min}, v_{\max}]$, so no approximation error arises. As the policy $\pi_k$ is progressively optimized, the Q-values increase, causing the target $\mathcal{T}^{\pi_k} Q$ to grow until it eventually exceeds $[v_{\min}, v_{\max}]$, leading to a significant projection error. In response, AHL-Gaussian dynamically expands $[v_{\min}, v_{\max}]$ to reduce the projection error. As a result, $[v_{\min}, v_{\max}]$ grows dynamically to track the increase in the true $Q^{\pi_k}$, thereby alleviating the approximation error.
> > >
> > > We sincerely hope that our further explanation clarifies our theoretical contributions, allowing you to reassess our work.

---

### Official Review · Reviewer_KKWb · 2024-11-03

**Soundness:** 4
**Presentation:** 3
**Contribution:** 3
**Rating:** 5
**Confidence:** 3

**Summary:**

The paper introduces Adaptive HL-Gaussian (AHL-Gaussian), an approach to learn categorical distributions for value functions in reinforcement learning. AHL-Gaussian improves upon the previous HL-Gaussian approach, which uses a fixed interval for the categorical distribution's support. The primary limitation of HL-Gaussian is that a poorly chosen interval can restrict the value function to an inaccurate range, leading to significant truncation and discretization errors.

To address this, the authors conduct a theoretical analysis of how interval selection affects error rates, clarifying the failure cases of the standard HL-Gaussian. Based on these insights, they propose AHL-Gaussian, which dynamically adjusts the support interval to align with the evolving value function. Experimental results demonstrate that AHL-Gaussian outperforms the static approach, advancing state-of-the-art performance.

**Strengths:**

The main strength of the paper is the theoretical analysis of the dependency between projection error and the support interval. The derivations are easy to follow and are supported by explanatory figures. The section concludes with a very clear and concise summary of the main findings.

The experimental section is also well-structured and the experiments' goals are clearly explained. The authors use Atari and Mujoco and apply AHL-Gaussian to DQN, TD3, and SAC.

**Weaknesses:**

In my eyes, the experiments are not fully convincing. I.e., the improvement over other methods seems to be environment dependent and pretty minor overall. Besides that, I doubt that the comparison is fair, as all baselines use fixed interval [-10, 10] for all environments.

- It is unclear why the interval of [-10, 10] is used for all non-adaptive methods for all environments. It looks like an unfair comparison to me. I would want to see how AHL-Gaussian compares to baselines with more adequate intervals.

- In Section 4.3, auhtors compare their method against non-learning approaches for updating the interval. As $1.1 * \text{Max Target}$ performs better than $1 * \text{Max Target}$, I would expect to see experiments with larger values. Besides, it seems natural to test other non-learning approaches. e.g., setting interval as $[\frac{r_{min}}{1-\gamma},\frac{r_{max}}{1-\gamma}]$, where $r_{min}, r_{max}$ are min/max rewards observed during training.

**Questions:**

My main question, as explained, is related to practical usability of the method. For which problems do the authors recommend using AHL-Gaussian as opposed to treating the support interval as a tunable hyperparameter?

Typos:
Line 486: Typo ``TTo’’.

Line 711: Missing $\mathbb{E}$

---

> ### Author Response · Authors · 2024-11-16
> **Response to Reviewer KKWb**
>
> Thank you very much for your questions and suggestions. We will respond one by one as follows:
>
> **1、 About the interval choice for baselines**
>
> First, we would like to clarify a misunderstanding by the reviewer regarding the interval selection for the baseline algorithms. In fact, we adopted different interval-setting strategies for Atari and Gym Mujoco environments:
>
> For the Atari environment, since the rewards are normalized to $[-1,1]$ and  [1] recommends a fixed interval of $[-10,10]$ as the optimal hyperparameter, we chose $[-10,10]$ as the hyperparameter for the baseline.
>
> For the Gym Mujoco environment, we fine-tuned an optimal interval for each task. The fine-tuning process is described in detail in the last paragraph of Section C in the appendix of the paper. Of course, limited hyperparameter search cannot guarantee the discovery of the true optimal hyperparameter. This thus highlights one of the advantages of our method: it eliminates the need for hyperparameter tuning and is robust to hyperparameter settings.
>
> [1] Stop regressing: Training value functions via classification for scalable deep rl.
>
>
> **2、 About other non-learning baselines**
>
> We would like to thank the reviewer for the suggestions. In response, we have added several non-learning baselines for further comparison in Section F.
>
> **Method 1**: The interval bound $\xi$ is set as the maximum value of all current Bellman targets, multiplied by a larger coefficient $\eta = 2$.
>
> From Figure 17, it is evident that this approach performs comparably to AHL-Gaussian on certain tasks. However, it exhibits significant shortcomings in tasks such as Swimmer and HumanoidStandup. Specifically, the target values in the Swimmer task fluctuate drastically, making this "overly sensitive" adjustment method unable to converge to a reasonable interval, which severely impacts training performance.
>
> **Method 2**: The support interval is set to $[\frac{r_{\min}}{1-\gamma}, \frac{r_{\max}}{1-\gamma}]$, where $r_{\min}$ and $r_{\max}$ are observed during training.
>
> As shown in Figure 18, this method also demonstrates disadvantages in tasks such as Swimmer and HumanoidStandup. This is likely  because it relies on discovering effective rewards during training, rendering it unsuitable for tasks with highly fluctuating reward signals or those that are challenging to explore.
>
> **Method 3**: This method normarlizes the reward and sets the support interval at $[-\frac{1}{1-\gamma}, \frac{1}{1-\gamma}]$.
>
> Figure  19 shows that this method performs significantly worse than AHL-Gaussian across multiple tasks. This is likely because directly setting the support interval to $1/(1-\gamma)$ causes unnecessarily large projection errors during the early stages of training, when the received reward values are relatively small compared to the interval range. This negatively affects training performance. Moreover, obtaining a reliable prior estimate of the maximum reward is challenging in practice.
>
> In summary, while these non-learning-based approaches achieve good performance on certain tasks, they have inherent limitations that make them unsuitable for a wide range of tasks. In contrast, our approach offers a more general and formal framework.
>
> **3、What types of problems is AHL-Gaussian more suitable for?**
>
> As outlined above, AHL-Gaussian may surpass parameter fine-tuning methods in scenarios characterized by highly fluctuating reward signals, challenging exploration, and limited prior knowledge.

---

> > ### Comment · Reviewer_KKWb · 2024-11-26
> >
> > Thank you for addressing my questions and concerns. However, I would still like to keep my score.
> >
> > The paper:
> > a) Provides theoretical bounds on the projection error (PE).
> > b) Tunes the support interval by performing gradient descent on the PE.
> >
> > While I think point (a) is a valuable contribution, I find (b) (subjectively) less compelling.
> > The experiments demonstrate that AHL-Gaussian, on average, performs slightly better than non-learning/non-adaptive baselines, though with high variance across problems and methods. Therefore, I am not convinced it can serve as a substitute for problem-specific fine-tuning of the support interval.

---

> ### Author Response · Authors · 2024-11-25
> **Looking forward to Continuing the Discussion**
>
> Dear Reviewer KKWb:
>
> We sincerely thank you for your efforts and valuable feedback on reviewing our paper. We noticed that you have not yet participated in the discussion, and we would like to ask if you have any additional comments or questions that we can address collaboratively. We have revised the manuscript based on your insightful feedback and are eager to address any remaining concerns.
>
> Please let us know if you have any further thoughts or questions. We look forward to continuing our discussion.
>
> Best regards,
>
> Authors

---

> ### Author Response · Authors · 2024-11-27
> **Response to Reviewer KKWb**
>
> Thank you very much for taking the time and effort to read our response and provide new comments. We appreciate your recognition of our theoretical contributions, and we would like to offer further clarification regarding our experimental contributions.
>
> * First, your acknowledgment that "AHL-Gaussian, on average, performs slightly better than non-learning/non-adaptive baselines" indicates that our method has good generalizability and consistently outperforms many alternatives.
>
> * Additionally, you expressed uncertainty about whether "it can serve as a substitute for problem-specific fine-tuning of the support interval." In fact, the experimental results on Gym Mujoco tasks have clearly shown that our approach is **superior to baselines with fine-tuned support intervals**.
>
> * Furthermore, we believe that if an algorithm requires task-specific fine-tuning of hyperparameters, its scalability and generalization ability in practical applications are limited. In contrast, our approach is robust to such task-dependent hyperparameter sensitivity, allowing it to achieve strong performance with any initial support interval, regardless of the task. This practical advantage is also one of the key experimental contributions of our method.
>
> We hope our reply addresses your concerns and please let us know if you have any further thoughts or questions.

---

### Official Review · Reviewer_egeR · 2024-11-09

**Soundness:** 3
**Presentation:** 3
**Contribution:** 3
**Rating:** 6
**Confidence:** 3

**Summary:**

This paper introduces a novel adaptive method for value based RL using cross entropy loss with discretized value space. Prior works have shown that using cross entropy loss instead of mean squared error improves the performance of RL algorithms but it also leads to suboptimal performance if value space is poorly discretized. The authors show that static discretization in HL-Gaussian leads to a projection error which increases with increase in interval size. The then propose an adaptive method that dynamically adjusts the support interval of value function during training by minimizing the projection error. They conduct extensive experiments to demonstrate the efficacy of their method over other baselines on various benchmark environments.

**Strengths:**

1. The paper theoretically characterizes the deficiencies of the standard HL Gaussian method based on projection error.

2. It proposes a novel adaptive algorithm that dynamically adapts the value range and corresponding discretization by minimizing the projection error.

3. Their method can be integrated with various RL baselines like DQN, SAC, TD3 and it shows nearly consistent improvement in performance when compared to standard RL training.

**Weaknesses:**

1. The proposed algorithm lacks theoretical insights on why quantization should help in training. There are no theoretical guarantee on performance of the algorithm.
2. The experiments are mostly conducted on simple RL environments and it is not yet clear if the method would work well on more complex real world environments like Dota, Minecraft etc.

**Questions:**

1. How does the choice of bin quantization(number and varying size) affect the performance of the algorithm? Why not adaptively optimize them as well?

---

> ### Author Response · Authors · 2024-11-16
> **Response to Reviewer egeR**
>
> Thank you very much for your questions and suggestions. We will respond to each of them as follows:
>
> **1、About the theoretical insights on why quantization should help in training**
>
> First, as the primary handling technique in distributional RL methods, **quantization** has been extensively discussed in C51 [1] for its benefits during training. Specifically, by learning the distribution of Q values instead of merely predicting the scalar Q values, it can better handle approximation errors, reduce chattering caused by policy updates, and mitigate state aliasing, thus improving training stability. Additionally, the distribution itself provides a rich set of predictions, allowing the agent to learn from multiple predictions rather than solely focusing on an expected value. Moreover, the distributional perspective introduces a more natural inductive bias framework for reinforcement learning, enabling the imposition of assumptions on the domain or the learning problem itself. This discussion has been included in the appendix of the revision.
>
> Second, **quantization** changes the optimization objective from an MSE loss to a cross-entropy loss (CE loss). The advantages of the HL-Gaussian-based CE loss used in this paper have been explained in Lines 157-161 and Lines 176-183, namely: *CE loss facilitates a more efficient path to the optimal solution with a reduced number of gradient steps*. The related theoretical analysis can be found in [2].
>
> In summary, **quantization** contributes to training through two dimensions: robust representation and efficient optimization.
>
> However, the aforementioned advantages do not account for the projection error introduced by mapping the Bellman target onto the categorical distribution due to an inappropriate interval range. In fact, as demonstrated in Proposition 3.1 of the main text, projection error affects policy evaluation. Furthermore, we have added a section in the appendix discussing how projection error impacts the algorithm's final performance. These findings suggest that while **quantization** offers stable representations and efficient optimization, the projection error in its application remains overlooked, leaving room for further improvement.
>
> To address this, this paper aims to reduce projection error by dynamically adjusting the interval range, enabling **quantization** to achieve better practical outcomes.
>
> [1]  distributional Perspective on Reinforcement Learning.
>
> [2] Investigating the Histogram Loss in Regression.
>
> **2、About the theoretical guarantee on performance of the algorithm**
>
> We have added Section B.3 to discuss the impact of projection errors in HL-Gaussian on the final performance of the algorithm. In particular, Theorem B.1 reveals a linear relationship between the projection error during each policy evaluation step and the final suboptimality gap, highlighting the importance of reducing this error.
>
> **3、About more complex tasks**
>
> We have added experiments on two complex and sophisticated control tasks DMControl Figner-Spin and Fish-Swim to showcase the advantages of AHL-Gaussian, please refer to Section G in the appendix for more details.
>
> **4、How does the choice of bin quantization(number and varying size) affect the performance of the algorithm? Why not adaptively optimize them as well?**
>
> * In term of the number of bins $m$：
>
> First, we have already demonstrated the impact of $m$ on AHL-Gaussian in Figure 8. Figure 8 shows that AHL-Gaussian generally performs robustly regardless of $m$, although there is a slight performance drop on a few tasks when $m$ is set too low.
>
> Furthermore, adaptively optimizing $m$ during training is not practical, as the output of the Q-network is $m$-dimensional. If $m$ changes dynamically, the structure of the neural network would also need to change dynamically, which would undoubtedly pose significant challenges for both training and inference.
>
> * In term of varying size：
>
> We are not entirely sure what the reviewer means by "varying size". We speculate that you might be referring to the bin width or the support interval range.
>
> In fact, when the number of bins $m$ is given, the bin width and the interval range are equivalent. We have already verified in Figure 1 that the performance of HL-Gaussian varies significantly under different interval ranges. Based on this observation, in this paper we proposed the dynamic optimization of the interval range (i.e., bin width) to address the differing requirements of interval ranges across environments and the dynamic evolving process of the value function.

---

> ### Author Response · Authors · 2024-11-25
> **Looking forward to Continuing the Discussion**
>
> Dear Reviewer egeR:
>
> We sincerely thank you for your efforts and valuable feedback on reviewing our paper. We noticed that you have not yet participated in the discussion, and we would like to ask if you have any additional comments or questions that we can address collaboratively. We have revised the manuscript based on your insightful feedback and are eager to address any remaining concerns.
>
> Please let us know if you have any further thoughts or questions. We look forward to continuing our discussion.
>
> Best regards,
>
> Authors

---

> > ### Comment · Reviewer_egeR · 2024-11-26
> >
> > Thank you for the replies! I have gone through your response and would like to maintain my score.

---

> > > ### Author Response · Authors · 2024-11-27
> > > **Response to Reviewer egeR**
> > >
> > > Thank you very much for taking the time and effort to review our manuscript and read our response, and we also appreciate your positive feedback on the paper.

---

### Author Response · Authors · 2024-11-16
**Brief Introduction to the Revisison**

First, we would like to thank the reviewers for taking the time to review our paper and provide valuable feedback. These suggestions have greatly contributed to improving the quality of our work. Below, we briefly summarize the changes made in the revision, followed by individual responses to each reviewer’s comments:

1. We have added Section B.3 in the appendix to discuss the cumulative effect of the projection error generated by the HL-Gaussian algorithm on the final performance. This section emphasizes the importance of reducing the projection error during each policy evaluation step.

2. We have introduced three new non-learning-based methods as baselines in Section F of the appendix to further compare with AHL-Gaussian. This comparison highlights the versatility and consistency of AHL-Gaussian. Additionally, in Section G, we have added experiments on two more complex control tasks to showcase the advantages of AHL-Gaussian.

3. We have included training curves for $\xi$ and projection error in Section E of the appendix, which correspond to the main experimental results. These curves help visualize the dynamic process of these variables.

4. We have corrected typos and clarified points where the reviewers had misunderstandings, both in the main text and the appendix.

All modifications and new additions have been highlighted in blue.

---

### Author Response · Authors · 2024-11-24
**Looking forward to Continuing the Discussion**

Dear Reviewers,

We hope this message finds you well. Thank you once again for your time and thoughtful feedback on our paper. We deeply value the opportunity to engage with you during the rebuttal phase, as your insights are instrumental in improving the quality of our work.

As the discussion phase approaches its deadline, we wanted to kindly check if there are any additional comments or questions you might have. We would greatly appreciate the chance to address any remaining concerns or explore further clarifications.

Please don’t hesitate to share your thoughts—we remain eager to collaborate in refining the paper.

Thank you for your support, and we look forward to hearing from you.

Best regards,
The Authors

---

### Meta-Review · Area_Chair_yDBr · 2024-12-23

**Metareview:**

Thank you for your submission to ICLR. This paper presents an adaptive method for value function learning, called AHL-Gaussian, which extends the HL-Gaussian method by introducing a dynamic support interval for the categorical distribution of the value function.

This is a borderline submission. Reviewers agree that this paper provides a good characterization of deficiencies in the typical HL-Gaussian method, proposes a novel algorithm, and shows a thorough set of empirical results. However, the reviewers also had a number of concerns. In particular, some of the reviewers were not convinced of the scope and significance of the work, in comparison with simpler, non-adaptive baselines (especially when factoring in the differences in empirical performance between these methods, in certain cases). In addition, there were also concerns about the clarity of presentation in the connections between the theoretical results and the presented algorithm, which led to disagreement about the soundness of the theory. By the end of the rebuttal, a majority of the authors were unconvinced and had remaining concerns. Therefore, I recommend this paper for rejection.

**Additional Comments On Reviewer Discussion:**

During the response period, there was a healthy discussion between the author and multiple reviewers. Though, by the end of rebuttal, reviewers still had concerns about the scope and significance of this work, and about whether some empirical results support the paper’s claims. There were additional questions remaining about the connection between the theoretical claims and presented algorithm.

---

### Decision · Program_Chairs · 2025-01-22

Reject